# Fixed-Budget Best-Arm Identification in Sparse Linear Bandits

**Recep Can Yavas**                                                   *recep.yavas@cnrsatcreate.sg*
*CNRS at CREATE, Singapore*

**Vincent Y. F. Tan**                                                          *vtan@nus.edu.sg*
*Department of Mathematics,*
*Department of Electrical and Computer Engineering,*
*National University of Singapore*

**Reviewed on OpenReview:** *https://openreview.net/forum?id=Igxp7FC8uf*

## Abstract

We study the best-arm identification problem in sparse linear bandits under the fixed-budget setting. In sparse linear bandits, the unknown feature vector $\boldsymbol{\theta}^*$ may be of large dimension $d$, but only a few, say $s \ll d$ of these features have non-zero values. We design a two-phase algorithm, Lasso and Optimal-Design- (Lasso-OD) based linear best-arm identification. The first phase of Lasso-OD leverages the sparsity of the feature vector by applying the thresholded Lasso introduced by Zhou (2009), which estimates the support of $\boldsymbol{\theta}^*$ correctly with high probability using rewards from the selected arms and a judicious choice of the design matrix. The second phase of Lasso-OD applies the OD-LinBAI algorithm by Yang and Tan (2022) on that estimated support. We derive a non-asymptotic upper bound on the error probability of Lasso-OD by carefully choosing hyperparameters (such as Lasso's regularization parameter) and balancing the error probabilities of both phases. For fixed sparsity $s$ and budget $T$, the exponent in the error probability of Lasso-OD depends on $s$ but not on the dimension $d$, yielding a significant performance improvement for sparse and high-dimensional linear bandits. Furthermore, we show that Lasso-OD is almost minimax optimal in the exponent. Finally, we provide numerical examples to demonstrate the significant performance improvement over the existing algorithms for non-sparse linear bandits such as OD-LinBAI, BayesGap, Peace, LinearExploration, and GSE.

## 1 Introduction

The stochastic multi-armed bandit (MAB) is a model that provides a mathematical formulation to study the sequential design of experiments and exploration-exploitation trade-off, where a learner pulls an arm out of a total $K$ and receives a reward drawn from a fixed and unknown distribution according to the chosen arm. This model has several applications including online advertising, recommendation systems, and drug tests. While in the standard reward model, the arms are uncorrelated with each other, stochastic linear bandits introduced in Auer (2002) generalize the standard model by associating each arm with a $d$-dimensional feature vector and the reward is equal to the inner product between the feature vector and an unknown global parameter. Therefore, the arms are correlated in linear bandits, meaning that pulling an arm gives information about the rewards of some other arms.

Most prior work including Auer (2002); Thompson (1933); Robbins (1952); Bubeck & Cesa-Bianchi (2012); Dani et al. (2008) on MABs focuses on *regret minimization*, where the goal is to maximize the cumulative reward after $T$ arm pulls by optimizing the trade-off between exploration and exploitation. Recently, the *pure exploration* setting has drawn attention from researchers. One example of pure exploration is the *best-arm identification* (BAI) problem, where the goal is to identify the arm with the largest mean reward. The

BAI problem is studied in two settings: (1) the fixed-budget setting considers a budget $T \in \mathbb{N}$ and aims to minimize the probability of failing to identify the best arm in at most $T$ arm pulls; (2) the fixed-confidence setting considers a confidence level $\delta \in (0,1)$ and aims to minimize the average number of arm pulls while identifying the best arm with probability at least $1 - \delta$.

For the standard reward model with uncorrelated arms, the works in Even-Dar et al. (2006); Karnin et al. (2013); Kaufmann et al. (2016) and Carpentier & Locatelli (2016); Audibert & Bubeck (2010) consider the BAI problem in the fixed-confidence and fixed-budget settings, respectively. For the linear model, the works in Soare et al. (2014); Xu et al. (2018); Fiez et al. (2019); Tao et al. (2018); Jedra & Proutiere (2020); Zaki et al. (2022) develop several algorithms under the fixed-confidence setting. For the linear model under the fixed-budget setting, Hoffman et al. (2014) develop the first algorithm, BayesGap, which is a gap-based exploration algorithm using a Bayesian approach. Katz-Samuels et al. (2020) develop the Peace algorithm that has equally-sized rounds, where the arm-pulling strategy within each round is based on the Gaussian width of the underlying arm set. Alieva et al. (2021) develop LinearExploration that exploits the linear structure of the model and is robust to unknown levels of observation noise and misspecification in the linear model. Yang & Tan (2022) develop the Optimal-Design-Based Linear Best Arm Identification (OD-LinBAI) algorithm, which also employs almost equally-sized rounds, but the arm-pulling strategy within each round is based on the G-optimal design. In the first round, OD-LinBAI aggressively eliminates all empirically suboptimal arms except the top $\frac{d}{2}$ arms; in the subsequent rounds, half of the remaining arms are eliminated in each round until a single arm remains. Azizi et al. (2022) develop the Generalized Successive Elimination (GSE) algorithm that has similar principles as OD-LinBAI with the difference that GSE eliminates the half of the remaining arms in all rounds. Among these algorithms, only OD-LinBAI is shown to be asymptotically minimax optimal.

In many practical applications of MABs, there are a large number of features available to the learner, but only a few of these features significantly affect the value of the reward of an arm. Sparse linear bandits are a mathematical abstraction of this phenomenon by assuming that the $d$-dimensional unknown parameter $\boldsymbol{\theta}^*$ in the linear model has only $s$ nonzero values, i.e., $\|\boldsymbol{\theta}^*\|_0 = s$, where $s$ is usually much smaller than $d$. The performance in the MAB problems (e.g., cumulative regret, probability of identification error) usually deteriorates as the ambient dimension $d$ increases. Therefore, the goal in the sparse setting is to design an algorithm whose performance is a function of $s$ but not $d$. Some works that study the regret minimization problem for sparse linear bandits include Abbasi-Yadkori et al. (2012); Kim & Paik (2019); Hao et al. (2020); Oh et al. (2021); Ariu et al. (2022); Li et al. (2022); Jang et al. (2022); Wang et al. (2023); Chakraborty et al. (2023). The OFUL algorithm of Abbasi-Yadkori et al. (2012) keeps track of a high probability confidence set for $\boldsymbol{\theta}^*$ and pulls an arm that maximizes the reward with respect to the arm vectors and the confidence set for $\boldsymbol{\theta}^*$. The DR Lasso algorithm of Kim & Paik (2019) combines Lasso with a doubly-robust technique used in the missing data literature. The ESTC algorithm of Hao et al. (2020) uses Lasso to estimate $\boldsymbol{\theta}^*$ at the end of the first phase and then in the second phase commits to the best arm with respect to the Lasso estimate. The SA Lasso Bandit algorithm of Oh et al. (2021) estimates $\boldsymbol{\theta}^*$ at each time using Lasso and pulls the best arm with respect to the Lasso estimate. TH Lasso Bandit algorithm of Ariu et al. (2022) estimates the support of $\boldsymbol{\theta}^*$ using Lasso and a thresholding procedure at each time and pulls the best arm with respect to the ordinary least squares estimation restricted to the estimated support in the first phase. Li et al. (2022) generalize the ESTC algorithm of Hao et al. (2020) to general bandit problems with low-dimensional structures such as low-rank matrix bandits. The PopArt algorithm of Jang et al. (2022) takes the population covariance of arms as input and uses a thresholding step to estimate $\boldsymbol{\theta}^*$ in the first phase; in the second phase, it commits to the best arm with respect to the estimate of $\boldsymbol{\theta}^*$ in the first phase. The LRP-Bandit algorithm of Wang et al. (2023) combines the thresholded Lasso with random projection where random projection is used to mitigate the negative influence of model misspecification due to the Lasso phase; their algorithm is also computationally efficient since Lasso is computed only at times with exponentially increasing gaps. Finally, Chakraborty et al. (2023) develop a Thompson Sampling algorithm for sparse linear contextual bandits.

In this paper, we study the BAI problem in sparse linear bandits under the fixed-budget setting. To the best of our knowledge, this paper presents the first result on the BAI problem in linear bandits with sparse structure, and we show that our bound on the error probability is almost minimax optimal in the exponent.

**Contributions**   Our main contributions are summarized as follows.

1. We design an algorithm, *Lasso and Optimal-Design- (Lasso-OD) based Linear Best Arm Identification.* This algorithm has two phases. In the first phase, we pull arms to estimate a support set $\hat{\mathcal{S}}$ that captures the support of the unknown parameter $\boldsymbol{\theta}^*$ with high probability and has size as small as possible. This goal is accomplished by the thresholded Lasso (TL) introduced by Zhou (2009). TL obtains an initial estimation $\hat{\boldsymbol{\theta}}_{\text{init}}$ for the parameter $\boldsymbol{\theta}^*$ from Lasso (Tibshirani, 1996) and passes it through an absolute value threshold to obtain $\hat{\boldsymbol{\theta}}_{\text{thres}}$. The support of $\hat{\boldsymbol{\theta}}_{\text{thres}}$ is the output of the first phase. In the second phase, we apply OD-LinBAI from Yang & Tan (2022). Lasso-OD has 3 hyperparameters: (i) $T_1 < T$, the budget allocated for the first phase; (ii) $\lambda_{\text{init}} > 0$, the parameter in the initial Lasso problem; and (iii) $\lambda_{\text{thres}} > 0$, the threshold value in TL. The choice of the design matrix (i.e., number of times each arm is pulled) in the first phase is crucial in attaining a good performance. Inspired by Hao et al. (2020), we design it by maximizing the smallest eigenvalue of the Gram matrix associated with the design matrix; this is known as the *E-optimal design* (Boyd & Vandenberghe, 2004, Sec. 7.5.2). This particular choice minimizes an upper bound on a probability term related to the performance of TL.

2. We derive a non-asymptotic upper bound on the error probability of Lasso-OD as a function of the total budget $T$, the number of arms $K$, the ambient dimension $d$, the sparsity $s$, and the arm vectors $\boldsymbol{a}(k)$, $k = 1, \ldots, K$, the first few suboptimality gaps, and the hyperparameters $T_1, \lambda_{\text{init}}$, and $\lambda_{\text{thres}}$. As a corollary to this bound, with the knowledge of $s$, we carefully choose the hyperparameters so that firstly, with high probability, phase 1 selects all variables in $\boldsymbol{\theta}^*$ and at most $s^2$ additional variables and secondly, the probability terms due to phases 1 and 2 are approximately "balanced". Under the assumption that the *compatibility constant*, which is a quantity that governs the performance of the Lasso, is lower bounded by a constant that is independent of $s$ and $d$, our particular choice achieves the error probability $\exp\left\{-\Omega\left(\frac{T}{(\log_2 s)H_{2,\text{lin}}(s+s^2)}\right)\right\}$ for $s, d, T \to \infty$, $\frac{s}{d} \to 0$, and $K$ and $d$ not growing exponentially with $T$ (see Corollary 1). Here, $H_{2,\text{lin}}(s + s^2)$ is a hardness parameter that depends only on the first $s + s^2 - 1$ suboptimality gaps. Note that the exponent is independent of dimension $d$, implying that increase in $d$ does not significantly increase the error probability. For OD-LinBAI, this exponent is given by $\exp\left\{-\Omega\left(\frac{T}{(\log_2 d)H_{2,\text{lin}}(d)}\right)\right\}$; therefore, Lasso-OD improves the error probability exponent by a factor of $\Omega\left(\frac{\log_2 d}{\log_2 s}\right)$ for $d \geq s + s^2$.

3. We empirically compare the identification error of Lasso-OD with that of other existing algorithms in the literature on several synthetic datasets, including one that is a sparsity-based version of examples used in other papers (Jedra & Proutiere, 2020). The empirical results support our theoretical result that claims that the scaling of the error probability of Lasso-OD is characterized by the sparsity $s$ while the performances of other algorithms significantly depend on $d$. We additionally demonstrate that algorithms that do not exploit the sparsity of $\boldsymbol{\theta}^*$ are computationally prohibitive, while the computational complexity of Lasso-OD scales well even as $d$ grows.

## 2   Problem Formulation

We consider a standard linear bandit with $K$ arms with a $d$-dimensional unknown global parameter $\boldsymbol{\theta}^*$. Let the arm set be $[K] \triangleq \{1, \ldots, K\}$, where each arm $k \in [K]$ is associated with a known arm vector $\boldsymbol{a}(k) \in \mathbb{R}^d$. A set of $K$ arms, $\{\boldsymbol{a}(1), \ldots, \boldsymbol{a}(k)\}$, together with $\boldsymbol{\theta}^*$ define a linear bandit instance $\eta$. At each time $t$, the agent chooses an arm $A_t \in [K]$ and observes a noisy reward

$$\text{y}_t = \langle \boldsymbol{\theta}^*, \boldsymbol{a}(A_t) \rangle + \epsilon_t, \tag{1}$$

where $\epsilon_1, \epsilon_2, \ldots$ are independent 1-subgaussian noise variables. For the arm selection, the agent uses an online algorithm, that is, the arm pull $A_t \in [K]$ may depend only on the previous $t - 1$ arm pulls $A_1, \ldots, A_{t-1}$ and their corresponding rewards $\text{y}_1, \ldots, \text{y}_{t-1}$. Denote the mean rewards of the arm vectors by

$$\mu_k \triangleq \langle \boldsymbol{\theta}^*, \boldsymbol{a}(k) \rangle, \quad \forall k \in [K]. \tag{2}$$

Without loss of generality, we assume that $\mu_1 > \mu_2 \geq \mu_3 \geq \cdots \geq \mu_K$, i.e., arm 1 is the unique best arm. We denote the mean gaps by $\Delta_k \triangleq \mu_1 - \mu_k$ for $2 \leq k \leq K$.

Under the fixed-budget setting of BAI, the agent is given a fixed time $T$, and makes an estimate $\hat{I}$ for the best arm with no more than $T$ arm pulls. The goal is to design an online algorithm with the identification error probability, $\mathbb{P}[\hat{I} \neq 1]$, as small as possible.

**Notation:** For any integer $n$, we denote $[n] \triangleq \{1, \ldots, n\}$. Let $\boldsymbol{x} = (x_1, \ldots, x_d)$ be a $d$-dimensional vector and $\mathcal{S} \subseteq [d]$, we denote $\boldsymbol{x}_{\mathcal{S}} \triangleq (x_s : s \in \mathcal{S}) \in \mathbb{R}^{|\mathcal{S}|}$. We denote $\|\boldsymbol{x}\|_{\boldsymbol{A}} \triangleq \sqrt{\boldsymbol{x}^\top \boldsymbol{A} \boldsymbol{x}}$. The minimum eigenvalue of a symmetric $\boldsymbol{A}$ is denoted by $\sigma_{\min}(\boldsymbol{A})$. We denote the set of distributions on the set $\mathcal{A}$ as $\mathcal{P}(\mathcal{A})$. Let $A_1, \ldots, A_t \in [K]$ be a sequence of arm pulls. The matrix $\boldsymbol{X} \in \mathbb{R}^{t \times d}$ whose $j$-th row is $\boldsymbol{a}(A_j)^\top$ is called the *design matrix*. Let $\nu \in \mathcal{P}([K])$ be the vector of fractions of arm pulls associated with this strategy, i.e., $\nu_k = \frac{1}{t} \sum_{j=1}^t \mathbb{1}\{A_j = k\}$ for $k \in [K]$. The *Gram matrix* associated with this strategy is denoted by $\boldsymbol{M}(\nu) = \frac{1}{t} \boldsymbol{X}^\top \boldsymbol{X} = \sum_{k \in [K]} \nu_k \boldsymbol{a}(k) \boldsymbol{a}(k)^\top \in \mathbb{R}^{d \times d}$. When we use asymptotic notation such as $O(\cdot)$ and $\Omega(\cdot)$, somewhat unconventionally, we are referring to *nonnegative* sequences, e.g., $a_n \in O(b_n)$ if and only if $\limsup_{n \to \infty} \frac{a_n}{b_n} < \infty$ and $\{a_n\}_{n \geq 1}$ is a nonnegative sequence.

**Model assumptions:** Denote the support of $\boldsymbol{\theta}^*$ by $S(\boldsymbol{\theta}^*) \triangleq \{j \in [d] : \theta_j^* \neq 0\}$. We assume that the unknown parameter $\boldsymbol{\theta}^*$ and the arm vectors $\{\boldsymbol{a}(k)\}_{k \in [K]}$ are of length $d$ but $\boldsymbol{\theta}^*$ is sparse, i.e., the number of non-zero coefficients in $\boldsymbol{\theta}^*$ satisfies $\|\boldsymbol{\theta}^*\|_0 \triangleq |S(\boldsymbol{\theta}^*)| = s < d$. We assume that $S(\boldsymbol{\theta}^*)$ is unknown, but $s$ and $\theta_{\min} \triangleq \min_{j \in S(\boldsymbol{\theta}^*)} |\theta_j^*|$ are known. We further assume that $|\mu_k| \leq 1$ for all arms $k \in [K]$ and that there exists a positive constant $\theta_0$ independent of $s$ and $d$ such that $\theta_{\min} \geq \theta_0$.

## 3    Our Algorithm: Lasso-OD

We now present our algorithm, *Lasso and Optimal-Design- (Lasso-OD) based linear best-arm identification* which has two phases. In phase 1, we pull a judiciously chosen set of arms to learn the support of the unknown parameter $\boldsymbol{\theta}^*$. Specifically, we design phase 1 so that it outputs a subset of variables $\hat{\mathcal{S}} \subseteq [d]$ whose support $\hat{\mathcal{S}}$ captures the true variables, $S(\boldsymbol{\theta}^*)$, with high probability, and its cardinality $|\hat{\mathcal{S}}|$ is small. To do this, we use the thresholded Lasso introduced by Zhou (2009). Once $\hat{\mathcal{S}}$ is obtained, we eliminate all variables in the arm vectors except the ones in $\hat{\mathcal{S}}$. Note that given that $\hat{\mathcal{S}} \supseteq S(\boldsymbol{\theta}^*)$, this variable elimination would have no effect on the mean values $\mu_1, \ldots, \mu_K$ since by assumption, we only eliminate some variables $j \in [d]$ with $\theta_j^* = 0$. Therefore, the best arm is also preserved after variable elimination. Building upon this principle, in phase 2, we project the arms on the estimated support $\hat{\mathcal{S}}$ and pull arms according to the OD-LinBAI algorithm by Yang & Tan (2022), which is designed for linear bandits without the sparsity assumption.

### 3.1    Motivation for Lasso-OD Algorithm

OD-LinBAI used in phase 2 is a minimax optimal algorithm up to a multiplicative factor in the exponent in the sense that it achieves an asymptotic error probability $\exp\left\{-\Omega\left(\frac{T}{(\log_2 d) H_{2,\text{lin}}(d)}\right)\right\}$, and for every algorithm, there exists a bandit instance $\eta$ whose asymptotic error probability is lower bounded by $\exp\left\{-O\left(\frac{T}{(\log_2 d) H_{2,\text{lin}}(d)}\right)\right\}$. The hardness parameter

$$H_{2,\text{lin}}(d) \triangleq \max_{2 \leq i \leq d} \frac{i}{\Delta_i^2} \tag{3}$$

determines how difficult it is to identify the best arm for a given bandit instance $\eta$ (Yang & Tan, 2022). For sparse linear bandits, if an oracle knew the support of the unknown parameter $\boldsymbol{\theta}^*$, then the lower bound in Yang & Tan (2022, Th. 3) would be improved to $\exp\left\{-O\left(\frac{T}{(\log_2 s) H_{2,\text{lin}}(s)}\right)\right\}$. The purpose of TL in phase 1 is to provide an estimate for the support of $\boldsymbol{\theta}^*$ with high accuracy while also pulling arms few enough that the resulting error probability is a function of $s$ rather than $d$ as in the oracle lower bound. Below, we provide the details on two phases of Lasso-OD.

### 3.2 Phase 1 (TL)

Consider a linear model $\mathbf{y} = \boldsymbol{X}\boldsymbol{\theta}^* + \boldsymbol{\epsilon}$, where $\boldsymbol{X} \in \mathbb{R}^{T_1 \times d}$ is a fixed design matrix, $\boldsymbol{\theta}^* \in \mathbb{R}^d$ is a fixed unknown feature vector, $\mathbf{y} \in \mathbb{R}^{T_1}$ is the response vector, and $\boldsymbol{\epsilon} \in \mathbb{R}^{T_1}$ is a noise vector whose entries are independent and 1-subgaussian. Tibshirani (1996) introduces the Lasso optimization problem to identify a sparse solution to the least squares estimation problem

$$\hat{\boldsymbol{\theta}}_{\text{init}} = \arg\min_{\boldsymbol{\theta} \in \mathbb{R}^d} \frac{1}{T_1} \|\mathbf{y} - \boldsymbol{X}\boldsymbol{\theta}\|_2^2 + \lambda_{\text{init}} \|\boldsymbol{\theta}\|_1, \tag{4}$$

where $\lambda_{\text{init}} > 0$ is a suitably chosen regularization parameter. The Lasso (4) is a convex program and can be solved efficiently, e.g., using Alternating Direction Method of Multipliers (ADMM) algorithm (Boyd et al., 2011).

For the task of variable selection, i.e., recovering the support of the unknown parameter $\boldsymbol{\theta}^*$ without missing any of its non-zero variables, we want to obtain an estimate $\hat{\boldsymbol{\theta}}$ that satisfies $S(\hat{\boldsymbol{\theta}}) \supseteq S(\boldsymbol{\theta}^*)$ while ensuring that $|S(\hat{\boldsymbol{\theta}}) \setminus S(\boldsymbol{\theta}^*)|$ is as small as possible. Zhou (2009) introduces the following thresholding procedure that has this property

$$(\hat{\boldsymbol{\theta}}_{\text{thres}})_j = (\hat{\boldsymbol{\theta}}_{\text{init}})_j \, \mathbb{1}\{|(\hat{\boldsymbol{\theta}}_{\text{init}})_j| \geq \lambda_{\text{thres}}\}, \quad \forall j \in [d], \tag{5}$$

where the initial estimate $\hat{\boldsymbol{\theta}}_{\text{init}}$ is given in (4), and $\lambda_{\text{thres}} > 0$ is the threshold. The set of selected variables by TL is $S(\hat{\boldsymbol{\theta}}_{\text{thres}})$. A variation of TL is used by Ariu et al. (2022) to derive refined regret guarantees in sparse stochastic contextual linear bandits. Their main idea is to find the support estimate $S(\hat{\boldsymbol{\theta}}_{\text{thres}}^{(t)})$ at each time instance $t$ using TL and then to compute the ordinary least squares (OLS) estimation restricted on the variables in $S(\hat{\boldsymbol{\theta}}_{\text{thres}}^{(t)})$. Ariu et al. (2022) tune the free parameters $\lambda_{\text{init}}^{(t)}$ and $\lambda_{\text{thres}}^{(t)}$ in a way that with high probability, $S(\hat{\boldsymbol{\theta}}_{\text{thres}}^{(t)}) \supseteq S(\boldsymbol{\theta}^*)$ and $S(\hat{\boldsymbol{\theta}}_{\text{thres}}^{(t)})$ is small enough, which is $s + O(\sqrt{s})$ in their case. Note that on the event $\{S(\hat{\boldsymbol{\theta}}_{\text{thres}}^{(t)}) \supseteq S(\boldsymbol{\theta}^*)\}$, the OLS solution restricted on the subset $S(\hat{\boldsymbol{\theta}}_{\text{thres}}^{(t)})$ is equal to that for the unrestricted case where all $d$ variables are used. Our approach is similar to that in Ariu et al. (2022) in using TL to reduce the effective dimension of the problem.

Let $T_1 < T$ be the budget allocated to the variable selection procedure described above.

**Design matrix optimization**  First, we need to specify the number of pulls for each arm during phase 1, which corresponds to determining the design matrix $\boldsymbol{X} \in \mathbb{R}^{T_1 \times d}$ in the Lasso problem (4). To do this, we solve the optimization problem, known as the *E-optimal design* (Boyd & Vandenberghe, 2004, Sec. 7.5.2), given by

$$\tilde{\nu}^{\star} = \arg\max_{\nu \in \mathcal{P}([K])} \sigma_{\min} \left( \sum_{i=1}^{K} \nu_i \boldsymbol{a}(i)\boldsymbol{a}(i)^{\top} \right). \tag{6}$$

Since the function $\boldsymbol{A} \mapsto \sigma_{\min}(\boldsymbol{A})$ is concave and $\nu \mapsto \sum_{i=1}^{K} \nu_i \boldsymbol{a}(i)\boldsymbol{a}(i)^{\top}$ is linear, (6) is a convex optimization problem, and can be solved efficiently, for example, using the CVX toolbox (Boyd et al., 2011).

The design matrix determined by the allocation in (6) minimizes an upper bound on a probability term related to phase 1; hence, it approximately optimizes the penalty term due to incorrectly estimating the variables of $\boldsymbol{\theta}^*$. More discussion on this choice of the design matrix appears in Appendix A. The optimization problem (6) also appears in Hao et al. (2020) on their regret analysis in sparse linear bandits. The allocation $\tilde{\nu}^{\star}$ can lead to fractional number of pulls $T_1 \tilde{\nu}_i^{\star}$ for some arm $i \in [K]$. To guarantee integer number of pulls for all arms, we apply a rounding procedure given in Pukelsheim (2006, Ch. 12), the ROUND function in Appendix B, which is also employed in the fixed-confidence BAI algorithm in Fiez et al. (2019).

**Support estimation**  We compute the number of pulls for each arm using (6) and ROUND, and then estimate the support from (4) and (5). Algorithm 1 below delineates the pseudo-code of this procedure.

---

**Algorithm 1** Thresholded Lasso (TL)

---

**input** Time budget $T_1$, Lasso parameters $\lambda_{\text{init}}$ and $\lambda_{\text{thres}}$, and arm vectors $\boldsymbol{a}(1), \ldots, \boldsymbol{a}(K)$.

 1: Compute the arm pull fractions $\tilde{\nu}^*$ from (6).
 2: Update $\tilde{\nu}^* \leftarrow \text{ROUND}(\tilde{\nu}^*, T_1)$ to ensure integer number of arm pulls.
 3: Pull each arm $i \in [K]$ exactly $T_1 \tilde{\nu}_i^*$ times. Denote the vector of rewards by $\mathbf{y} \in \mathbb{R}^{T_1}$.
 4: Form the design matrix $\boldsymbol{X} \in \mathbb{R}^{T_1 \times d}$ so that it has $T_1 \tilde{\nu}_i^*$ rows equal to $\boldsymbol{a}(i)^\top$ for $i \in [K]$. Compute $\hat{\boldsymbol{\theta}}_{\text{thres}}$ from (4) and (5).

**output** the support $\hat{\mathcal{S}} = S(\hat{\boldsymbol{\theta}}_{\text{thres}})$.

---

### 3.3 Phase 2 (OD-LinBAI)

In this section, we review the OD-LinBAI algorithm by Yang & Tan (2022). OD-LinBAI divides the budget $T$ into $\lceil \log_2 d \rceil$ phases, where each phase has roughly the same length.

At the start of round $r$, OD-LinBAI applies a dimensionality reduction step to maintain that the set of modified arms spans the space of its reduced dimension. The arm allocation during each round is determined by the *G-optimal design* (Kiefer & Wolfowitz, 1960), which takes a set of arm vectors $\{\boldsymbol{a}(1), \ldots, \boldsymbol{a}(K)\} \subseteq \mathbb{R}^d$ and solves the optimization problem

$$\pi^* = \underset{\pi \in \mathcal{P}([K])}{\arg\min} \max_{i \in [K]} \|\boldsymbol{a}(i)\|_{\boldsymbol{M}(\pi)^{-1}}^2 , \tag{7}$$

where $\boldsymbol{M}(\pi) \triangleq \sum_{i=1}^{K} \pi_i \boldsymbol{a}(i) \boldsymbol{a}(i)^\top$ is the Gram matrix associated with the allocation $\pi$. At the start of each round, we solve (7) for the set of active arms and then apply the ROUND function in Appendix B to the resulting allocation to ensure integer number of pulls. The latter step replaces the procedure in Line 17 Yang & Tan (2022, Algorithm 1). This slight modification may improve the performance of the algorithm especially if the budget $T$ is small. At the end of round 1, we eliminate all arms except the top $\lceil \frac{d}{2} \rceil$ with respect to the OLS estimator; in the rest, we halve the remaining arms at the end each round. At the end of the last round, only one arm remains and that arm is declared to be the best one. The pseudo-code of OD-LinBAI can be found in Yang & Tan (2022) and a slight modification of it which leads to the improved error probability bound in Theorem 2 can be found in Appendix B.

### 3.4 Lasso-OD Algorithm

The pseudo-code of Lasso-OD described above is given in Algorithm 2. Notice that since the two phases of Lasso-OD operate independently, one can replace either or both of TL and OD-LinBAI with their alternatives, e.g., the PopArt algorithm (Jang et al., 2022) and the adaptive Lasso (Bühlmann & van de Geer, 2011, Ch. 2.8) for TL and any of the algorithms in Alieva et al. (2021); Katz-Samuels et al. (2020); Azizi et al. (2022); Hoffman et al. (2014) for OD-LinBAI. We discuss some of the variants of our algorithm in Appendix G.

---

**Algorithm 2** Lasso and Optimal-Design Based Linear Best Arm Identification (Lasso-OD)

---

**input** Time budgets $T_1$ and $T_2$ so that $T = T_1 + T_2$, Lasso parameters $\lambda_{\text{init}}$ and $\lambda_{\text{thres}}$, and arm vectors $\boldsymbol{a}(1), \ldots, \boldsymbol{a}(K) \in \mathbb{R}^d$.

 1: Run TL (Algorithm 1) with $T_1, \lambda_{\text{init}}$, and $\lambda_{\text{thres}}$ and get the output $\hat{\mathcal{S}} \subseteq [d]$.
 2: Project the arm vectors on the subset $\hat{\mathcal{S}}$ by setting $\boldsymbol{a}'(i) = (\boldsymbol{a}(i))_{\hat{\mathcal{S}}}$ for $i \in [K]$.
 3: Run OD-LinBAI from Yang & Tan (2022) with budget $T_2$ and arm vectors $\{\boldsymbol{a}'(1), \ldots, \boldsymbol{a}'(K)\} \subseteq \mathbb{R}^{|\hat{\mathcal{S}}|}$ with Line 17 of Algorithm 1 in Yang & Tan (2022) replaced by ROUND.

**output** the only remaining arm $\hat{I}$ as the output of OD-LinBAI.

---

## 4 Main Results

This section presents three non-asymptotic upper bounds on the performances of TL, OD-LinBAI, and Lasso-OD algorithms.

### 4.1 Thresholded Lasso

Recall the linear model $\mathbf{y} = \boldsymbol{X}\boldsymbol{\theta}^* + \boldsymbol{\epsilon}$, where $\boldsymbol{X} \in \mathbb{R}^{T_1 \times d}$ is a fixed design matrix, $\boldsymbol{\theta}^* \in \mathbb{R}^d$ is a fixed unknown feature vector, $\mathbf{y} \in \mathbb{R}^{T_1}$ is the response vector, and $\boldsymbol{\epsilon} \in \mathbb{R}^{T_1}$ is a noise vector whose entries are independent and 1-subgaussian. For any set $\mathcal{S} \subseteq [d]$, define the set of vectors

$$\mathbb{C}(\mathcal{S}) \triangleq \{\boldsymbol{\theta} \in \mathbb{R}^d \colon \|\boldsymbol{\theta}_{\mathcal{S}^c}\|_1 \leq 3 \|\boldsymbol{\theta}_{\mathcal{S}}\|_1\}. \tag{8}$$

van de Geer & Bühlmann (2009) introduce the following *compatibility condition* that allows one to control the $\ell_1$-norm error for the sparse estimation of the unknown parameter $\boldsymbol{\theta}^*$ where the components of the design matrix $\boldsymbol{X}$ are not highly correlated. For the rest of the section, let $\boldsymbol{M} = \frac{1}{T_1}\boldsymbol{X}^\top\boldsymbol{X}$ denote the Gram matrix associated with $\boldsymbol{X}$.

**Definition 1** (Compatibility condition)**.** *Given a fixed design matrix $\boldsymbol{X} \in \mathbb{R}^{T_1 \times d}$ (whose Gram matrix is $\boldsymbol{M}$) and a subset $\mathcal{S} \subseteq [d]$, the compatibility constant $\phi^2(\boldsymbol{M}, \mathcal{S})$ is defined as*

$$\phi^2(\boldsymbol{M}, \mathcal{S}) \triangleq \min_{\boldsymbol{\theta} \in \mathbb{R}^d \colon \|\boldsymbol{\theta}_{\mathcal{S}}\|_1 \neq 0} \left\{ \frac{|\mathcal{S}| \|\boldsymbol{\theta}\|_{\boldsymbol{M}}^2}{\|\theta_{\mathcal{S}}\|_1^2} \colon \boldsymbol{\theta} \in \mathbb{C}(\mathcal{S}) \right\}. \tag{9}$$

*With some abuse of notation, we also define*

$$\phi^2(\boldsymbol{M}, s) \triangleq \min_{\mathcal{S} \subseteq [d] \colon |\mathcal{S}| = s} \phi^2(\boldsymbol{M}, \mathcal{S}). \tag{10}$$

The following result controls the $\ell_1$-norm error of the initial Lasso estimator in (4).

**Lemma 1** (Ariu et al. (2022), Lemma G.6)**.** *Assume that $\phi^2(\boldsymbol{M}, s) > 0$. The Lasso estimator $\hat{\boldsymbol{\theta}}_{\mathrm{init}}$ in (4) satisfies*

$$\mathbb{P}\left[\left\|\hat{\boldsymbol{\theta}}_{\mathrm{init}} - \boldsymbol{\theta}^*\right\|_1 \leq \frac{4\lambda_{\mathrm{init}}s}{\phi^2(\boldsymbol{M}, s)}\right] \geq 1 - 2d \exp\left\{-\frac{T_1\lambda_{\mathrm{init}}^2}{32\left(\frac{1}{T_1}\max_{j \in [d]}\|\boldsymbol{X}_{:,j}\|_2^2\right)}\right\}. \tag{11}$$

Using Lemma 1, we derive the following bound on the event that the size of the support of the TL output (5) is below a threshold and it captures the true support $S(\boldsymbol{\theta}^*)$.

**Theorem 1.** *Fix a design matrix $\boldsymbol{X} \in \mathbb{R}^{T_1 \times d}$ and parameters $\lambda_{\mathrm{init}}, \lambda_{\mathrm{thres}} > 0$. Let $c = \frac{\lambda_{\mathrm{thres}}}{\lambda_{\mathrm{init}}}$. Assume that $\phi^2 \triangleq \phi^2(\boldsymbol{M}, s) > 0$ and $\theta_{\min} \geq \lambda_{\mathrm{init}}\left(c + \frac{4}{\phi^2}s\right)$ holds. Then,*

$$\mathbb{P}\left[\left\{|S(\hat{\boldsymbol{\theta}}_{\mathrm{thres}})| \leq s\left(1 + \frac{4}{\phi^2 c}\right)\right\}\bigcap\{S(\hat{\boldsymbol{\theta}}_{\mathrm{thres}}) \supseteq S(\boldsymbol{\theta}^*)\}\right] \geq 1 - 2d \exp\left\{-\frac{T_1\lambda_{\mathrm{init}}^2}{32\left(\frac{1}{T_1}\max_{j \in [d]}\|\boldsymbol{X}_{:,j}\|_2^2\right)}\right\}. \tag{12}$$

The proofs of Lemma 1 and Theorem 1 are deferred to Appendix C. Theorem 1 follows steps similar to those in Ariu et al. (2022, Lemma 5.4). The interested reader can refer to Bühlmann & van de Geer (2011, Ch. 6 and 7) for more results and discussions on Lasso, TL, and their variants.

## 4.2 An Improved Upper Bound on the Error Probability of OD-LinBAI

The theorem below gives an improved upper bound on the error probability of OD-LinBAI (Yang & Tan, 2022).

**Theorem 2.** *Let $\tilde{T} = \left\lfloor \frac{T}{\lceil \log_2 d \rceil} \right\rfloor$. For any linear bandit instance, the output of OD-LinBAI satisfies*

$$\mathbb{P}\left[ \hat{I} \neq 1 \right] \leq (K + \log_2 d) \exp \left\{ -\frac{\tilde{T}}{16 \left( 1 + \frac{d^2}{\tilde{T}} \right) H_{2,\mathrm{lin}}(d)} \right\}. \tag{13}$$

The right-hand side of (13) is slightly different than the one presented in Yang & Tan (2022, Th. 2). First, in Yang & Tan (2022, Th. 2), the numerator in the exponent is equal to some constant $m$ that is approximately equal to $\frac{T}{\log_2 d}$ just like $\tilde{T}$; this is due to the modification in the distribution rounding technique. Second, the pre-factor in Yang & Tan (2022, Th. 2) is $\frac{4K}{d} + 3 \log_2 d$ instead of (our smaller) $K + \log_2 d$. More importantly, in (13), the constant 32 in the denominator of the exponent in Yang & Tan (2022, Th. 2) is improved to 16. The last two differences are due to a refinement in the proof technique. Lastly, our result includes a rounding error factor $1 + \frac{d^2}{\tilde{T}}$, which becomes negligible as $T$ becomes large. This factor appears due to the fact that the G-optimal design may yield fractional number of pulls for some arms, which is obviously not allowed in practice. The proof of Theorem 2 is deferred to Appendix D.

## 4.3 Upper Bound on the Error Probability of Lasso-OD

The theorem below bounds the probability of incorrectly identifying the best arm using Lasso-OD.

**Theorem 3.** *Let $T_1 < T$ be the length of phase 1, and let $T_2 = T - T_1$ be the length of phase 2. Let $\lambda_{\mathrm{init}}$ and $\lambda_{\mathrm{thres}}$ be some positive scalars. Let $c = \frac{\lambda_{\mathrm{thres}}}{\lambda_{\mathrm{init}}}$. Let $\tilde{\nu}^*$ be the solution to (6), and let $\tilde{\nu} = \mathrm{ROUND}(\tilde{\nu}^*, T_1)$ be its rounded version for length $T_1$. Assume that the compatibility constant associated with the E-optimal design is positive, i.e., $\phi^2 \triangleq \phi^2 \left( \sum_{i=1}^{K} \tilde{\nu}_i \boldsymbol{a}(i) \boldsymbol{a}(i)^\top, s \right) > 0$, and $\theta_{\min} \geq \lambda_{\mathrm{init}} \left( c + \frac{4}{\phi^2} s \right)$. Then, the output of Algorithm 2 satisfies*

$$\mathbb{P}\left[ \hat{I} \neq 1 \right] \leq (K + \log_2 d) \exp \left\{ -\frac{\left\lfloor \frac{T_2}{\log_2(s_1)} \right\rfloor}{16 \left( 1 + \epsilon \right) H_{2,\mathrm{lin}}(s_1)} \right\} + 2d \exp \left\{ -\frac{T_1 \lambda_{\mathrm{init}}^2}{32 x_{\max}^2} \right\}, \tag{14}$$

*where*

$$s_1 = \left\lfloor s \left( 1 + \frac{4}{\phi^2 c} \right) \right\rfloor, \quad x_{\max}^2 = \max_{j \in [d]} \sum_{k=1}^{K} \tilde{\nu}_k (\boldsymbol{a}(k)_j)^2, \quad and \quad \epsilon = \frac{s_1^2}{T_2}. \tag{15}$$

*Proof.* The proof uses Theorems 1 and 2 for the probability terms due TL and OD-LinBAI, respectively. Let $\hat{\mathcal{S}} \subseteq [d]$ denote the output of phase 1. Define the events $\mathcal{E} \triangleq \{|\hat{\mathcal{S}}| \leq s_1\}$ and $\mathcal{F} \triangleq \{\hat{\mathcal{S}} \supseteq S(\boldsymbol{\theta}^*)\}$. By the law of total probability, we have

$$\mathbb{P}\left[ \hat{I} \neq 1 \right] \leq \mathbb{P}\left[ \hat{I} \neq 1 \middle| \mathcal{E} \cap \mathcal{F} \right] + \mathbb{P}\left[ \mathcal{E}^{\mathrm{c}} \cup \mathcal{F}^{\mathrm{c}} \right]. \tag{16}$$

Given $\mathcal{E} \cap \mathcal{F}$, the error probability is bounded by the right-hand side of (13) with the budget $T$ replaced by the length of phase 2, $T_2$, and with the dimension $d$ replaced by $s_1$. This follows since on the event $\mathcal{F}$, the mean rewards are preserved after the arm vectors and $\boldsymbol{\theta}^*$ are projected on $\hat{\mathcal{S}}$ and since the right-hand side of (13) is non-decreasing in $d$. From Theorem 1 and the arm-pulling strategy described in Line 2 of Algorithm 2, we have

$$\mathbb{P}\left[ \mathcal{E}^{\mathrm{c}} \cup \mathcal{F}^{\mathrm{c}} \right] \leq 2d \exp \left\{ -\frac{T_1 \lambda_{\mathrm{init}}^2}{32 x_{\max}^2} \right\}. \tag{17}$$

Combining (16) with (13) and (17), we complete the proof. □

The following corollary is obtained by choosing the free parameters $T_1, \lambda_{\text{init}}$, and $\lambda_{\text{thres}}$ suitably to meet the conditions of Theorem 3. These nontrivial choices use the knowledge of $\theta_{\min}$ and $s$ but not the hardness parameter and achieve an exponent of the error probability that depends only on $s$, $T$, and the hardness parameter.

**Corollary 1.** *For any linear bandit instance that satisfies $\phi^2 > 0$, the output of Algorithm 2 satisfies*

$$\mathbb{P}\left[\hat{I} \neq 1\right] \leq (K + \log_2 d + 2d) \exp\left\{-\frac{T}{16\lfloor \log_2(s+s^2)\rfloor(1+\epsilon)H_{2,\text{lin}}(s+s^2)(1+c_0)}\right\}, \quad (18)$$

*where*

$$c_0 = \frac{400x_{\max}^2}{3\phi^4\theta_{\min}^2\log_2(s+s^2)} \quad and \quad \epsilon = \frac{(1+c_0)(s+s^2)^2}{T}. \quad (19)$$

Here, $c_0 = \frac{T_1}{T_2}$ is the fraction of lengths of two phases of Lasso-OD, and $1 + \epsilon$ is the small multiplicative penalty due to rounding.

Assume that for a sequence of bandit instances (of growing dimension $d$), there exist positive constants $\phi_0^2$ and $x_0^2$, both independent of $s$ and $d$, such that $\phi^2 \geq \phi_0^2$ and $x_{\max}^2 \leq x_0^2$.[1] A paradigmatic example of this is the scenario where the arm vectors are generated independently from a zero-mean, unit-variance distribution with a finite fourth-order moment (e.g., the Gaussian distribution $\mathcal{N}(0,1)$, equiprobable distribution on $\{-1,1\}$). Now, consider the setting in which $K = T_1 = rd$ where $r > 1$ is independent of $s$ and $d$, and we pull each arm once in the Lasso phase. Then, from Bai & Yin (1993, Theorem 2), we know that the minimum eigenvalue of the Gram matrix $\sigma_{\min}(\boldsymbol{M})$ converges to $\left(1 - \frac{1}{\sqrt{r}}\right)^2$ with probability one, and $x_{\max}^2$ converges to 1 also with probability one. For this example, we may take the constants $\phi_0^2$ and $x_0^2$ to be $\phi_0^2 = \frac{1}{2}\left(1 - \frac{1}{\sqrt{r}}\right)^2$ and $x_0^2 = 2$. Note that, for this example, $\phi_0^2$ and $x_0^2$ are independent of $s$ and $d$. Therefore, the ratio $c_0$ is $O(1)$ as $s$, $d$, and $T$ grow. Under these conditions, Corollary 1 implies that the error probability of Lasso-OD is upper bounded by

$$\exp\left\{-\Omega\left(\frac{T}{(\log_2 s)H_{2,\text{lin}}(s+s^2)}\right)\right\} \quad (20)$$

for $s, d, T \to \infty$, $\frac{s}{d} \to 0$, and $K$ and $d$ not growing exponentially with $T$, all of which are realistic assumptions in practice. Unlike the non-sparse case in Yang & Tan (2022), the error probability exponent under these assumptions is *independent of the dimension $d$*, but instead, depends on the sparsity $s$, which yields much smaller error probabilities for high dimensional sparse linear bandits. The choices of the parameters that achieve the exponent in (20) is nontrivial; we carefully choose $\lambda_{\text{init}}$ and $\lambda_{\text{thres}}$ so that the condition in Theorem 1 is satisfied with equality and $c_0$ is decreasing in $s$ and choose $T_1$ so that two exponents in (14) resulting from phases 1 and 2 are approximately equal. The proof of Corollary 1 is presented in Appendix E.

Assume that the agent knows the support of $\boldsymbol{\theta}^*$. Then, following the construction in the proof of Yang & Tan (2022, Th. 3), for any algorithm, there exists a bandit instance whose error probability is lower bounded by $\exp\left\{-O\left(\frac{T}{(\log_2 s)H_{2,\text{lin}}(s)}\right)\right\}$. This implies that the upper bound in (20) is indeed almost minimax optimal in the exponent. In Appendix G, we develop a variant of Lasso-OD, called PopArt-OD, which replaces TL in phase 1 of Lasso-OD with PopArt from Jang et al. (2022). Thanks to the fact that PopArt provides a guarantee on the $\ell_\infty$ norm of the difference between the estimated parameter $\boldsymbol{\theta}'$ and $\boldsymbol{\theta}^*$, we derive an upper bound on the probability $\mathbb{P}\left[\hat{\mathcal{S}}_{\text{PA}} \neq S(\boldsymbol{\theta}^*)\right]$, where $\hat{\mathcal{S}}_{\text{PA}}$ denotes the estimated support using the PopArt algorithm. Using this bound, we show that the error probability of PopArt-OD is upper bounded by $\exp\left\{-\Omega\left(\frac{T}{(\log_2 s)H_{2,\text{lin}}(s)}\right)\right\}$, matching the lower bound up to a constant factor in the exponent. Due to its superior empirical performance over PopArt-OD, we focus on Lasso-OD in the paper.

---

[1]In general, $\phi^2$ depends on the geometry of the set of arm vectors. One can construct instances where $\phi^2$ vanishes as $s$ and $d$ grow.

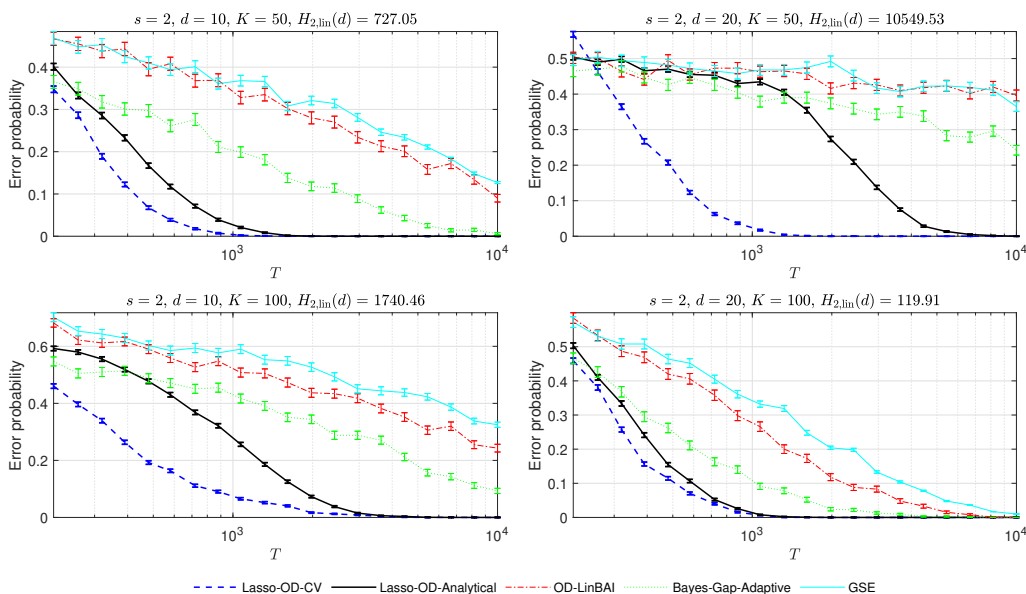

Figure 1: Comparison of several algorithms with $T \in [200, 10000]$ and $s = 2$.

Table 1: Performance comparison of several algorithms for $T = 800$, $d = 10$, $K = 50$, and $s = 2$.

|  | Lasso-OD-CV | Lasso-OD-An. | Peace | LinearExploration |
|---|---|---|---|---|
| Error probability | 0.0275 | 0.045 | 0.40 | 0.39 |
| Std. deviation | 0.0026 | 0.0033 | 0.049 | 0.0153 |

## 5 Experiments

In this section, we numerically evaluate the performance of Lasso-OD on several synthetic sparse linear bandit instances and compare it with those of OD-LinBAI (Yang & Tan, 2022), BayesGap (Hoffman et al., 2014), GSE (Azizi et al., 2022), Peace (Katz-Samuels et al., 2020), and LinearExploration (Alieva et al., 2021). In each setting, we report the empirical error probabilities for Lasso-OD, BayesGap, and GSE over 4000 independent trials and for Peace and LinearExploration over 100 independent trials.

### 5.1 Synthetic Dataset with Sparse Unknown Parameter Vector

To illustrate the efficacy and robustness of our algorithm on synthetic data, we consider three different sets of experiments.

In the first example, we draw $K$ arms independently from the uniform distribution on the $d$-dimensional sphere of radius $\sqrt{d/s}$, i.e., $\{x \in \mathbb{R}^d \colon \|x\|_2^2 = \frac{d}{s}\}$, and the sparse unknown parameter is taken as $\boldsymbol{\theta}^* = (1, 1, 0, \ldots, 0)$, i.e., $s = 2$. Figure 1 reports the empirical error probabilities for $d \in \{10, 20\}$, $K \in \{50, 100\}$ and $T \in [200, 10000]$, except Peace (Katz-Samuels et al., 2020) and LinearExploration (Alieva et al., 2021). Since the computational complexities of Peace and LinearExploration are much higher than the rest of the algorithms, we compare Lasso-OD with Peace and LinearExploration only for $T = 800$ in Table 1. Among these algorithms, Lasso-OD has the best performance for all sparse instances shown in Figure 1 and Table 1.

Lasso-OD-CV sets the budgets for phase 1 and phase 2 as $T_1 = \frac{T}{5}$ and $T_2 = \frac{4T}{5}$ and tunes the Lasso parameters $\lambda_{\text{init}}$ and $\lambda_{\text{thres}}$ using a $K$-fold cross-validation procedure that uses the value of $s$ in its loss function. See Appendix F for the details of the cross-validation procedure. As an alternative to cross-validation, Lasso-OD-Analytical uses the knowledge of $s$, $\theta_{\min}$, and the hardness parameter $H_{2,\text{lin}}(s_1)$ in (14),

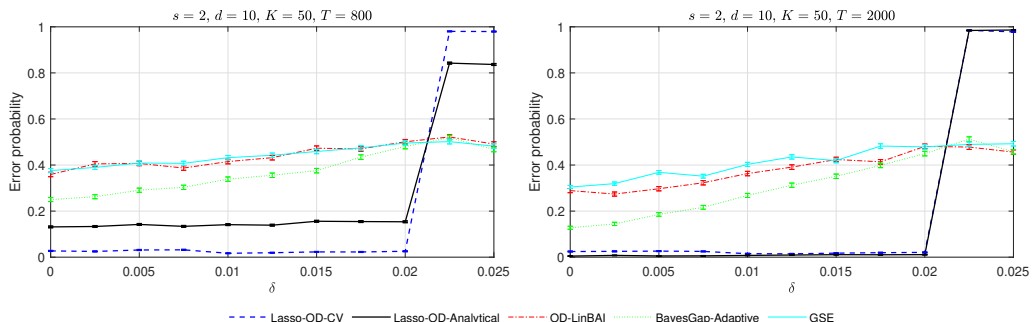

Figure 2: Comparison of several algorithms with $T \in \{800, 2000\}$, $s = 2$, and $\delta \in [0, 0.025]$.

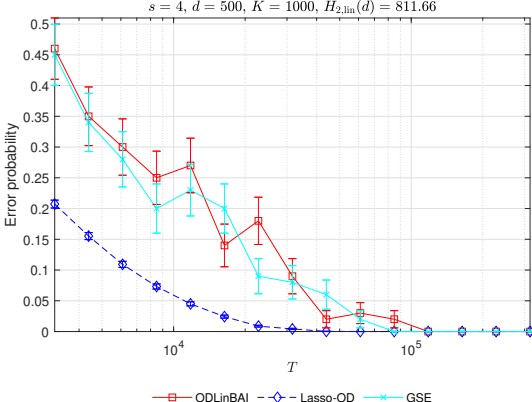

Figure 3: Comparison of Lasso-OD, OD-LinBAI, and GSE for $d = 500$, $s = 4$, and $T \in [3 \times 10^3, 3 \times 10^5]$.

and sets $\lambda_{\text{init}}$, $\lambda_{\text{thres}}$, and $T_1$ so that $s_1$ in (15) equals $s + s^2$, $\theta_{\min} = \lambda_{\text{init}}(c + bs)$, and two exponents in (14) are equal. Note that $H_{2,\text{lin}}(s_1)$ is usually not available to the agent.

In the second example, we test the robustness of our algorithm with respect to the variables in $\boldsymbol{\theta}^*$ that are assumed to be zero by keeping the same arms as in the previous example and setting $\boldsymbol{\theta}^*$ as $\theta_j^* = 1$ for $j \in [2]$, and $\theta_j^* = \delta R_j$ for $j \in \{3, \ldots, d\}$, where $R_j$, $j = 3, \ldots, d$, are independent Rademacher (i.e., $\{\pm 1\}$-valued) random variables, and $\delta > 0$ is a constant. Figure 2 reports the empirical error probabilities for $s = 2$, $d = 10$, $K = 50$, $T \in \{800, 2000\}$, and $\delta \in [0, 0.025]$. The phase transition for Lasso-OD in Figure 2 suggests that Lasso-OD achieves a smaller error probability as long as $\delta$ is small enough that the approximately sparse instance (i.e., $\delta > 0$) and the sparse instance (i.e., $\delta = 0$) have the same best arm. Some examples including an instance where the hyperparameters are set as in Corollary 1 without cross-validation or knowing the hardness parameter are discussed in Appendix G.

The third experiment is similar to the first, except that the parameters are larger to demonstrate that our method scales well, while others do not. Namely, we set $s = 4$, $d = 500$, and $K = 1000$. Since only Lasso-OD, OD-LinBAI, and GSE are the only computationally feasible algorithms for such large dimensions, we only include the performances of these algorithms in Figure 3. The empirical error probabilities are obtained from 4000 independent trials for Lasso-OD and from 100 independent trials for OD-LinBAI and GSE.[2] We observe that Lasso-OD outperforms OD-LinBAI for this instance at all $T$ values shown.

### 5.2 Real-World Dataset with Sparse Unknown Parameter Vector

We conduct an experiment on an online news popularity dataset published by Mashable (Fernandes et al., 2015), which includes 39,797 news articles, each having 58 attributes. Some of these attributes are the

---

[2]We use a smaller number of independent trials for OD-LinBAI and GSE as it takes too long to run since it does not exploit the sparsity of the problem.

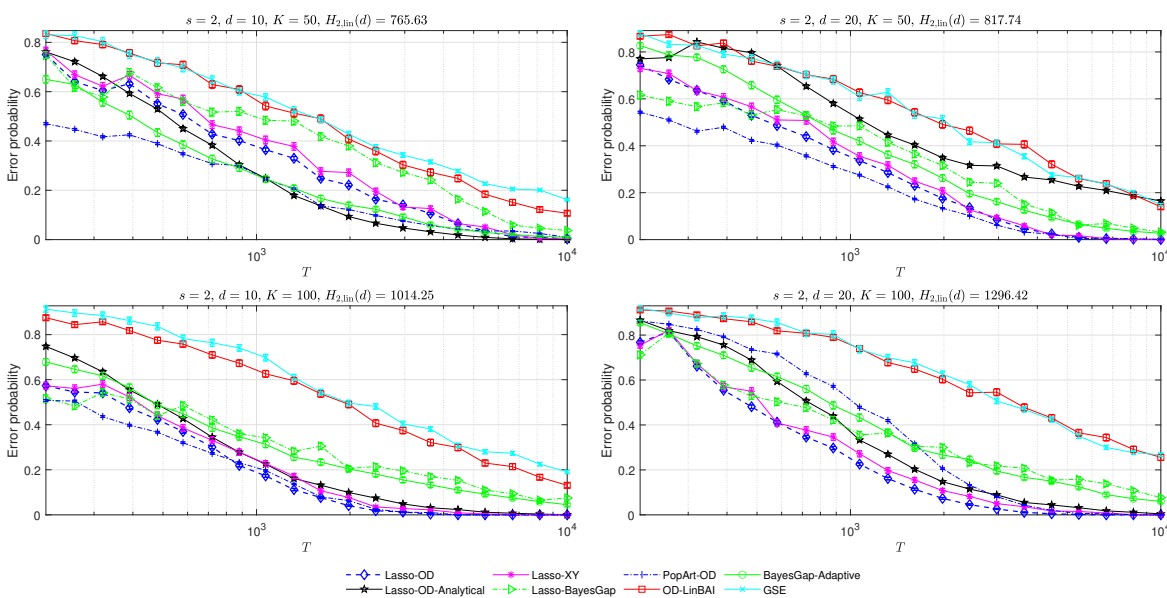

Figure 4: Comparison of several algorithms for the example bandit instance in Yang & Tan (2022).

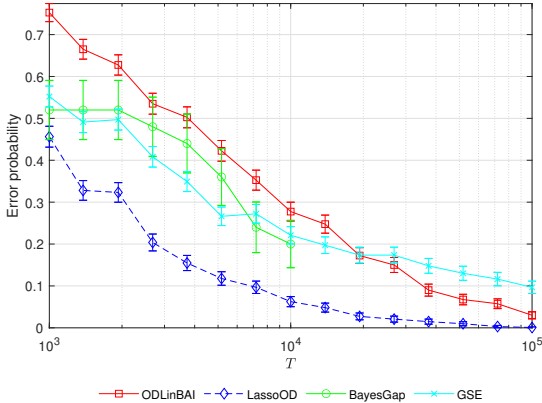

Figure 5: Comparison of several algorithms for the real-world dataset with $T \in [10^3, 10^5], s = 3$, and $d = 55$.

number of words, the number of links, the number of keywords, the day that the news is published, and the channel of the news. The target of the dataset is the number of shares in social network. To adapt the dataset to the sparse linear bandit framework, we first normalize each attribute (and the target) so that the mean and the standard deviation of each attribute is 0 and 1, respectively. Since three of the attributes have $> 99\%$ correlation with some other attribute, we remove these three attributes; hence the overall dimension $d = 55$. Then, to obtain a sparse ground truth vector $\boldsymbol{\theta}^* \in \mathbb{R}^{55}$ (assuming a linear model between the attributes and the target), we estimate $\boldsymbol{\theta}^*$ using the thresholded Lasso, where the hyperparameters $\lambda_{\text{init}}$ and $\lambda_{\text{thres}}$ are determined via cross-validation. The resulting $\boldsymbol{\theta}^*$ has 3 nonzero values, hence $s = 3$ (all 3 active attributes are related to the number of keywords in the news). We select 500 news articles with the largest shares as the arms, i.e., $K = 500$.

In Figure 5, we see that Lasso-OD, where its two hyperparameters $\lambda_{\text{init}}$ and $\lambda_{\text{thres}}$ are determined by cross-validation, has the smallest error probability for all $T$ values shown. We are not able to conduct experiments using Peace and LinearExploration due to their prohibitively high computational complexities for such large dimensions. To corroborate this claim, the CPU runtimes of the algorithms compared in the experiments

Table 2: The empirical means of the CPU runtimes for $d \in \{10, 55, 500\}$ and $T = 10^4$.

| | CPU runtimes (seconds) | | | | | |
|---|---|---|---|---|---|---|
| $d$ | Lasso-OD | OD-LinBAI | GSE | BayesGap-Ad. | LinearExploration | Peace |
| 10 | 0.0019 | 0.0084 | 0.011 | 0.75 | 2.33 | 31.69 |
| 55 | 0.0068 | 0.048 | 0.092 | 83.2 | >1800 | >1800 |
| 500 | 0.017 | 4.95 | 5.29 | >1800 | >1800 | >1800 |

above are shown in Table 2.[3] In all experiments, Lasso-OD is the fastest algorithm, and the gap between the CPU runtimes of the Lasso-OD and of the other algorithms increases with the dimension $d$, as the other algorithms do not exploit the sparsity of the unknown parameter vector $\boldsymbol{\theta}^*$.

# 6 Conclusion

In this work, we study the BAI problem in linear bandits with sparse structure under fixed-budget setting and develop the first BAI algorithm, Lasso-OD, that exploits the sparsity of the unknown parameter $\boldsymbol{\theta}^*$. Lasso-OD combines TL for support estimation with the minimax optimal BAI algorithm, OD-LinBAI. We analyze the error probability of Lasso-OD and show that the error exponent depends on the sparsity $s$ rather than the dimension $d$. Unlike other algorithms in the literature, the empirical performance of Lasso-OD does not deteriorate at large dimensions.

One future direction is to derive an instance-dependent asymptotic or non-asymptotic lower bound for the BAI problem in sparse linear bandits; however, such a bound remains open even in the non-sparse scenario. Another possible direction is to extend the TL technique used in Lasso-OD to the fixed-confidence setting. Although such an extension is relatively easy to analyze, the empirical performances of most fixed-confidence BAI algorithms in linear bandits are not heavily dependent on the dimension unlike the fixed-budget setting (see, for example, Zaki et al. (2022); Tao et al. (2018); Fiez et al. (2019)). Therefore, the benefit of adding a TL phase in the fixed-confidence setting could be limited.

# 7 Acknowledgments and Disclosure of Funding

This research is part of the programme DesCartes and is supported by the National Research Foundation, Prime Minister's Office, Singapore under its Campus for Research Excellence and Technological Enterprise (CREATE) programme.

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

# A   Design Matrix Optimization in Thresholded Lasso

The performance of TL in Lasso-OD is characterized by the bound on the probability in (12). For any pair of $(\lambda_{\text{init}}, \lambda_{\text{thres}})$, the bound in (12) is optimized by setting the design matrix $\boldsymbol{X}$ to maximize the ratio $\frac{\phi^2(\boldsymbol{M},s)}{\sqrt{\frac{1}{T_1}\max_{j\in[d]}\|\boldsymbol{X}_{:,j}\|_2}}$.

Note that in the application of the Lasso, we can normalize the dataset so that $\sum_{k\in[K]}\boldsymbol{a}(k)_j^2$ are equal for all $j \in [d]$. Under the assumptions that the number of arms, $K$, is large and that the arm vectors are drawn from the same distribution, the value of $\frac{1}{T_1}\max_{j\in[d]}\|\boldsymbol{X}_{:,j}\|_2^2$ cannot vary too much with respect to different fraction of arm pulls. Therefore, we relax the quantity $\frac{1}{T_1}\max_{j\in[d]}\|\boldsymbol{X}_{:,j}\|_2^2$ in (12) by its upper bound $\max_{k\in[K]}\|\boldsymbol{a}(k)\|_\infty^2$. Then, Theorem 1 implies that the best choice of $\boldsymbol{X}$ maximizes the compatibility constant $\phi^2(\boldsymbol{M}, s)$.

**Computation of the compatibility constant**   Let $\nu \in \mathcal{P}_{T_1}([K])$ be the $T_1$-type distribution describing the fractions of the number of pulls for each arm. Then, $\boldsymbol{M} = \frac{1}{T_1}\boldsymbol{X}^\top\boldsymbol{X} = \sum_{i\in[K]}\nu_i\boldsymbol{a}(i)\boldsymbol{a}(i)^\top$. Rewriting the compatibility constant $\phi^2(\boldsymbol{M}, \mathcal{S})$ from Definition 1, with some overload of notation, we obtain

$$\phi^2(\nu, \mathcal{S}) \triangleq \phi^2(\boldsymbol{M}, \mathcal{S}) = \min_{\boldsymbol{\theta}\in\mathbb{R}^d}\left\{|\mathcal{S}|\,\|\boldsymbol{\theta}\|^2_{\sum_{i\in[K]}\nu_i\boldsymbol{a}(i)\boldsymbol{a}(i)^\top} : \|\boldsymbol{\theta}_\mathcal{S}\|_1 = 1, \|\boldsymbol{\theta}_{\mathcal{S}^c}\|_1 \le 3\right\} \tag{21}$$

$$\phi^2(\nu, s) \triangleq \min_{\mathcal{S}\subseteq[d]:\,|\mathcal{S}|=s}\phi^2(\nu, \mathcal{S}). \tag{22}$$

Given a fixed $\nu$, the program in (21) is non-convex due to the $\ell_1$-norm equality constraint; however, by introducing binary variables, it can be turned into a mixed-integer discipled convex program (MIDCP) and be efficiently solved using CVX toolbox (Grant & Boyd, 2012). If we relaxed the equality constraint to $\|\boldsymbol{\theta}_\mathcal{S}\|_1 \le 1$, then (21) would be a quadratic program (QP).

**Relaxing the optimization problem**   According to the arguments above, the optimization problem that we originally need to solve is

$$\nu^* = \underset{\nu\in\mathcal{P}_{T_1}([K])}{\arg\max}\ \phi^2(\nu, s), \tag{23}$$

which is computationally intractable since the maximization constraint makes it an integer program; and even if we relaxed it to allow fractional number of pulls, the program would involve $\binom{d}{s} \approx d^s$ MIDCPs in its constraints.

**Lemma 2.** *For any $\nu \in \mathcal{P}([K])$ and any $\mathcal{S} \subseteq [d]$, it holds that $\phi^2(\nu, \mathcal{S}) \ge \sigma_{\min}\left(\sum_{i=1}^K \nu_i\boldsymbol{a}(i)\boldsymbol{a}(i)^\top\right)$.*

Lemma 2 follows from $|\mathcal{S}|\,\|\boldsymbol{\theta}_\mathcal{S}\|_1^2 \le \|\boldsymbol{\theta}_\mathcal{S}\|_2^2 \le \|\boldsymbol{\theta}\|_2^2$ and relaxing the inequality constraint in (21).

Replacing $\phi^2(\nu, \mathcal{S})$ by its lower bound and allowing fractional number of pulls, we get the relaxed optimization problem in (6), which can be solved efficiently.

With the inclusion of $\frac{1}{T_1}\max_{j\in[d]}\|\boldsymbol{X}_{:,j}\|_2^2$, the optimization problem can be re-formulated as

$$\max_{\nu\in\mathcal{P}([K])}\frac{\sigma_{\min}\left(\sum_{i=1}^K \nu_i\boldsymbol{a}(i)\boldsymbol{a}(i)^\top\right)}{\sqrt{\max\left(\text{diag}\left(\sum_{i=1}^K \nu_i\boldsymbol{a}(i)\boldsymbol{a}(i)^\top\right)\right)}}, \tag{24}$$

which does not satisfy the MIDCP ruleset. In our experiments, we observe that the convergence of the numerical solution to the problem in (24) heavily depends on the initialization; therefore, (24) could not be considered as a reliable approach.

# B  Pseudo-codes of ROUND and OD-LinBAI

The pseudo-codes of the rounding procedure from Pukelsheim (2006, Ch. 12) that is used in Algorithm 1 and OD-Lasso from Yang & Tan (2022) are given below.

---

**Algorithm 3** ROUND($\pi, T$)

---

**input** a distribution $\pi$ on a set with cardinality $d$ and a positive integer $T$.
1:  Initialize $T_i = \lceil (T - \frac{d}{2})\pi_i \rceil$ for $i = 1, \ldots, d$.
2:  **while** $\sum_{i=1}^{d} T_i \neq T$ **do**
3:    **if** $\sum_{i=1}^{d} T_i < T$ **then**
4:      Set $j \leftarrow \arg\min_{i \in [d]} \frac{T_i}{\pi_i}$. Update $T_j \leftarrow T_j + 1$.
5:    **else if** $\sum_{i=1}^{d} T_i > T$ **then**
6:      Set $j \leftarrow \arg\max_{i \in [d]} \frac{T_i - 1}{\pi_i}$. Update $T_j \leftarrow T_j - 1$.
7:    **end if**
8:  **end while**
**output** Distribution $\tilde{\pi} = \left( \frac{T_1}{T}, \ldots, \frac{T_d}{T} \right)$.

---

---

**Algorithm 4** Optimal Design-Based Linear Best Arm Identification (OD-LinBAI)

---

**input** time budget $T$, arm set $\mathcal{A} = [K]$, and arm vectors $\{\boldsymbol{a}(1), \ldots, \boldsymbol{a}(K)\} \in \mathbb{R}^d$.
1:  Initialize $t_0 \leftarrow 0$, $\mathcal{A}_0 \leftarrow \mathcal{A}$, $d_0 \leftarrow d$. For each $i \in \mathcal{A}_0$, set $\boldsymbol{a}_0(i) = \boldsymbol{a}(i)$. Set $R = \lceil \log_2 d \rceil$, $T_r = \lfloor \frac{T}{R} \rfloor$ for $r = 1, \ldots, R-1$, and $T_R = T - \sum_{i=1}^{R-1} T_i$.
2:  **for** $r = 1$ **to** $R$ **do**
3:    \\ Dimensionality reduction:
4:    Set $\boldsymbol{X}$ so that its columns are $\{\boldsymbol{a}_{r-1}(i) \colon i \in \mathcal{A}_{r-1}\}$. Set $d_r \leftarrow \text{rank}(\boldsymbol{X})$. Set $\boldsymbol{a}_r(i) \leftarrow \boldsymbol{a}_{r-1}(i)$ for $i \in \mathcal{A}_{r-1}$.
5:    **if** $d_r < d_{r-1}$ **then**
6:      Find the singular value decomposition of $\boldsymbol{X} = \boldsymbol{U}\boldsymbol{D}\boldsymbol{V}^\top$, where $\boldsymbol{U} \in \mathbb{R}^{d_{r-1} \times d_r}$.
7:      Update $\boldsymbol{X} \leftarrow \boldsymbol{U}^\top \boldsymbol{X}$ and $\boldsymbol{a}_r(i) \leftarrow \boldsymbol{X}_i$ for $i \in \mathcal{A}_{r-1}$.
8:    **end if**
9:    \\ G-optimal design:
10:   Input the set $\{\boldsymbol{a}_r(i) \colon i \in \mathcal{A}_{r-1}\}$ to the G-optimal design, and set $\pi^{(r)}$ as the output of (7).
11:   Set $\tilde{\pi}^{(r)} = \text{ROUND}(\pi^{(r)}, T_r)$ from Algorithm 2.
12:   \\ Arm pulling:
13:   Pull each arm $i \in \mathcal{A}_{r-1}$ $T_r(i) = \tilde{\pi}_i^{(r)} T_r$ times, which determines $A_{t_{r-1}+1}, \ldots, A_{t_{r-1}+T_r}$. Observe the corresponding rewards $\mathsf{y}_{t_{r-1}+1}, \ldots, \mathsf{y}_{t_{r-1}+T_r}$.
14:   Compute the OLS estimator

$$\boldsymbol{V}^{(r)} = \sum_{i \in \mathcal{A}_{r-1}} T_r(i) \boldsymbol{a}_r(i) \boldsymbol{a}_r(i)^\top \tag{25}$$

$$\hat{\boldsymbol{\theta}}^{(r)} = {\boldsymbol{V}^{(r)}}^{-1} \sum_{t=t_{r-1}+1}^{t_{r-1}+T_r} \boldsymbol{a}_r(A_t) \mathsf{y}_t. \tag{26}$$

15:   \\ Arm elimination:
16:   Estimate the mean rewards for each $i \in \mathcal{A}_{r-1}$ as

$$\hat{\mu}_r(i) = \langle \hat{\boldsymbol{\theta}}^{(r)}, \boldsymbol{a}_r(i) \rangle. \tag{27}$$

    Set $\mathcal{A}_r \leftarrow$ the set of $\lceil \frac{d}{2^r} \rceil$ arms in $\mathcal{A}_{r-1}$ with the largest estimated mean rewards. Set $t_r \leftarrow t_{r-1} + T_r$.
17: **end for**
**output** $\hat{I} =$ the only remaining arm in $\mathcal{A}_R$.

---

## C  Proofs Related to Lasso

In the following, let $n$ be the number of samples. The linear model is given by $\mathbf{y} = \boldsymbol{X}\boldsymbol{\theta}^* + \boldsymbol{\epsilon}$, where $\mathbf{y} \in \mathbb{R}^n$ are the rewards, $\boldsymbol{X} \in \mathbb{R}^{n \times d}$ is the design matrix, and $\boldsymbol{\epsilon} \in \mathbb{R}^n$ are i.i.d. 1-subgaussian random variables. Recall the initial Lasso estimator

$$\hat{\boldsymbol{\theta}} = \underset{\boldsymbol{\theta} \in \mathbb{R}^d}{\arg\min} \frac{1}{n} \|\mathbf{y} - \boldsymbol{X}\boldsymbol{\theta}\|_2^2 + \lambda \|\boldsymbol{\theta}\|_1. \tag{28}$$

Define the event

$$\mathcal{T} = \left\{ \max_{j \in [d]} \frac{1}{n} |\boldsymbol{X}_{:,j}^\top \boldsymbol{\epsilon}| \le \frac{\lambda}{4} \right\}. \tag{29}$$

The following result, known as the oracle inequality, is the main tool to control the performance of the initial lasso estimator.

**Lemma 3** (Oracle Inequality: Theorem 6.1 from Bühlmann & van de Geer (2011))**.** *On the event $\mathcal{T}$, the initial Lasso estimator $\hat{\boldsymbol{\theta}}$ (28) satisfies*

$$\left\| \boldsymbol{X}(\hat{\boldsymbol{\theta}} - \boldsymbol{\theta}^*) \right\|_2^2 + \lambda \left\| \hat{\boldsymbol{\theta}} - \boldsymbol{\theta}^* \right\|_1 \le \frac{4\lambda^2 s}{\phi^2(\boldsymbol{M}, S(\boldsymbol{\theta}^*))}. \tag{30}$$

*Furthermore, it holds that*

$$\mathbb{P}\left[\mathcal{T}\right] \ge 1 - 2d \exp\left\{ -\frac{n\lambda^2}{32 \left( \frac{1}{n} \max_{j \in [d]} \|\boldsymbol{X}_{:,j}\|_2^2 \right)} \right\}. \tag{31}$$

*Proof of Lemma 3.* Since $\hat{\boldsymbol{\theta}}$ minimizes (28), we have

$$\frac{1}{n} \left\| \mathbf{y} - \boldsymbol{X}\hat{\boldsymbol{\theta}} \right\|_2^2 + \lambda \left\| \hat{\boldsymbol{\theta}} \right\|_1 \le \frac{1}{n} \|\mathbf{y} - \boldsymbol{X}\boldsymbol{\theta}^*\|_2^2 + \lambda \|\boldsymbol{\theta}^*\|_1. \tag{32}$$

Plugging $\mathbf{y} = \boldsymbol{X}\boldsymbol{\theta}^* + \boldsymbol{\epsilon}$ into (32), after some algebra, we get the basic inequality

$$\frac{1}{n} \left\| \boldsymbol{X}(\hat{\boldsymbol{\theta}} - \boldsymbol{\theta}^*) \right\|_2^2 + \lambda \left\| \hat{\boldsymbol{\theta}} \right\|_1 \le \frac{2}{n} \boldsymbol{\epsilon}^\top \boldsymbol{X}(\hat{\boldsymbol{\theta}} - \boldsymbol{\theta}^*) + \lambda \|\boldsymbol{\theta}^*\|_1. \tag{33}$$

Let $\tilde{\mathcal{T}}$ be the event

$$\tilde{\mathcal{T}} = \left\{ \max_{j \in [d]} \frac{2}{n} |\boldsymbol{\epsilon}^\top \boldsymbol{X}_{:,j}| \le \lambda_0 \right\}. \tag{34}$$

Then, on $\tilde{\mathcal{T}}$, we have using the Hölder inequality that

$$\frac{1}{n} \left\| \boldsymbol{X}(\hat{\boldsymbol{\theta}} - \boldsymbol{\theta}^*) \right\|_2^2 \le \lambda_0 \left\| \hat{\boldsymbol{\theta}} - \boldsymbol{\theta}^* \right\|_1 + \lambda \|\boldsymbol{\theta}^*\|_1 - \lambda \left\| \hat{\boldsymbol{\theta}} \right\|_1. \tag{35}$$

Let $\mathcal{S} = S(\boldsymbol{\theta}^*)$. By the triangle inequality, we have

$$\left\| \hat{\boldsymbol{\theta}} \right\|_1 = \left\| \hat{\boldsymbol{\theta}}_{\mathcal{S}} \right\|_1 + \left\| \hat{\boldsymbol{\theta}}_{\mathcal{S}^c} \right\|_1 \ge \|\boldsymbol{\theta}^*_{\mathcal{S}}\|_1 - \left\| \hat{\boldsymbol{\theta}}_{\mathcal{S}} - \boldsymbol{\theta}^*_{\mathcal{S}} \right\|_1 + \left\| \hat{\boldsymbol{\theta}}_{\mathcal{S}^c} \right\|_1. \tag{36}$$

Applying (36) to (35), we get

$$\frac{1}{n} \left\| \boldsymbol{X}(\hat{\boldsymbol{\theta}} - \boldsymbol{\theta}^*) \right\|_2^2 \le \lambda_0 \left( \left\| \hat{\boldsymbol{\theta}}_{\mathcal{S}} - \boldsymbol{\theta}^*_{\mathcal{S}} \right\|_1 + \left\| \hat{\boldsymbol{\theta}}_{\mathcal{S}^c} - \boldsymbol{\theta}^*_{\mathcal{S}^c} \right\|_1 \right)$$
$$+ \lambda \left( \|\boldsymbol{\theta}^*\|_1 - \|\boldsymbol{\theta}^*_{\mathcal{S}}\|_1 + \left\| \hat{\boldsymbol{\theta}}_{\mathcal{S}} - \boldsymbol{\theta}^*_{\mathcal{S}} \right\|_1 - \left\| \hat{\boldsymbol{\theta}}_{\mathcal{S}^c} \right\|_1 \right) \tag{37}$$
$$= (\lambda_0 + \lambda) \left\| \hat{\boldsymbol{\theta}}_{\mathcal{S}} - \boldsymbol{\theta}^*_{\mathcal{S}} \right\|_1 + (\lambda_0 - \lambda) \left\| \hat{\boldsymbol{\theta}}_{\mathcal{S}^c} - \boldsymbol{\theta}^*_{\mathcal{S}^c} \right\|_1, \tag{38}$$

where the last step uses the fact that $\boldsymbol{\theta}^*_{\mathcal{S}^c} = 0$. We set $\lambda_0 = \frac{\lambda}{2}$. Then, (38) implies that on the event $\mathcal{T}$,

$$\left\|\hat{\boldsymbol{\theta}}_{\mathcal{S}^c} - \boldsymbol{\theta}^*_{\mathcal{S}^c}\right\|_1 \leq 3 \left\|\hat{\boldsymbol{\theta}}_{\mathcal{S}} - \boldsymbol{\theta}^*_{\mathcal{S}}\right\|_1. \tag{39}$$

Therefore, $\hat{\boldsymbol{\theta}} - \boldsymbol{\theta}^* \in \mathbb{C}(\mathcal{S})$, and from the definition of compatibility constant in Definition 1, we have

$$\left\|\hat{\boldsymbol{\theta}}_{\mathcal{S}} - \boldsymbol{\theta}^*_{\mathcal{S}}\right\|_1 \leq \frac{\sqrt{s(\hat{\boldsymbol{\theta}} - \boldsymbol{\theta}^*)^\top \boldsymbol{X}^\top \boldsymbol{X}(\hat{\boldsymbol{\theta}} - \boldsymbol{\theta}^*)}}{\sqrt{n}\phi(\boldsymbol{M}, \mathcal{S})}. \tag{40}$$

We now continue with (38) with $\lambda_0 = \frac{\lambda}{2}$. We have

$$\frac{2}{n}\left\|\boldsymbol{X}(\hat{\boldsymbol{\theta}} - \boldsymbol{\theta}^*)\right\|_2^2 + \lambda\left\|\hat{\boldsymbol{\theta}} - \boldsymbol{\theta}^*\right\|_1 = \frac{2}{n}\left\|\boldsymbol{X}(\hat{\boldsymbol{\theta}} - \boldsymbol{\theta}^*)\right\|_2^2 + \lambda\left\|\hat{\boldsymbol{\theta}}_{\mathcal{S}} - \boldsymbol{\theta}^*_{\mathcal{S}}\right\|_1 + \lambda\left\|\hat{\boldsymbol{\theta}}_{\mathcal{S}^c} - \boldsymbol{\theta}^*_{\mathcal{S}^c}\right\|_1 \tag{41}$$

$$\leq (3\lambda + \lambda)\left\|\hat{\boldsymbol{\theta}}_{\mathcal{S}} - \boldsymbol{\theta}^*_{\mathcal{S}}\right\|_1 - \lambda\left\|\hat{\boldsymbol{\theta}}_{\mathcal{S}^c} - \boldsymbol{\theta}^*_{\mathcal{S}^c}\right\|_1 + \lambda\left\|\hat{\boldsymbol{\theta}}_{\mathcal{S}^c} - \boldsymbol{\theta}^*_{\mathcal{S}^c}\right\|_1 \tag{42}$$

$$= 4\lambda\left\|\hat{\boldsymbol{\theta}}_{\mathcal{S}} - \boldsymbol{\theta}^*_{\mathcal{S}}\right\|_1 \tag{43}$$

$$\leq \frac{4\lambda\sqrt{s}\left\|\boldsymbol{X}(\hat{\boldsymbol{\theta}} - \boldsymbol{\theta}^*)\right\|_2}{\sqrt{n}\phi(\boldsymbol{M}, \mathcal{S})} \tag{44}$$

$$\leq \frac{1}{n}\left\|\boldsymbol{X}(\hat{\boldsymbol{\theta}} - \boldsymbol{\theta}^*)\right\|_2^2 + \frac{4\lambda^2 s}{\phi^2(\boldsymbol{M}, \mathcal{S})}, \tag{45}$$

where (44) applies (40), and (45) applies the inequality $4uv \leq u^2 + 4v^2$ to (44). Inequality (45) completes the proof of (30).

Next, we upper bound the probability $\mathbb{P}\left[\tilde{\mathcal{T}}^c\right]$. We have

$$\mathbb{P}\left[\tilde{\mathcal{T}}^c\right] = \mathbb{P}\left[\bigcup_{j \in [d]}\left\{\frac{1}{n}|\boldsymbol{X}^\top_{:,j}\boldsymbol{\epsilon}| > \frac{\lambda}{4}\right\}\right] \tag{46}$$

$$\leq \sum_{j=1}^{d}\left(\mathbb{P}\left[\frac{1}{n}\boldsymbol{X}^\top_{:,j}\boldsymbol{\epsilon} > \frac{\lambda}{4}\right] + \mathbb{P}\left[-\frac{1}{n}\boldsymbol{X}^\top_{:,j}\boldsymbol{\epsilon} > \frac{\lambda}{4}\right]\right) \tag{47}$$

$$\leq 2\sum_{j=1}^{d}\exp\left\{-\frac{\lambda^2}{2 \cdot 4^2 \cdot \frac{1}{n^2}\left\|\boldsymbol{X}_{:,j}\right\|_2^2}\right\}, \tag{48}$$

where the last inequality follows since $\frac{1}{n}\boldsymbol{X}^\top_{:,j}\boldsymbol{\epsilon} - \frac{1}{n}\boldsymbol{X}^\top_{:,j}\boldsymbol{\epsilon}$ are subgaussian with variance proxy $\frac{1}{n^2}\left\|\boldsymbol{X}_{:,j}\right\|_2^2$ as $\epsilon_1, \ldots, \epsilon_n$ are independent 1-subgaussian random variables. Bounding each summand in (48) by the maximum of summands completes the proof of (31). $\square$

*Proof of Lemma 1.* The right-hand side of (30) depends on the unknown set $S(\boldsymbol{\theta}^*)$; however, we can further upper bound the right-hand side of (30) by replacing $\phi^2(\boldsymbol{M}, S(\boldsymbol{\theta}^*))$ by its lower bound $\phi^2(\boldsymbol{M}, s)$, which is computable using only $\boldsymbol{X}$ and $s$. Therefore, Lemma 1 is a corollary to Lemma 3. $\square$

*Proof of Theorem 1.* Define the event $\mathcal{G} \triangleq \left\{\left\|\hat{\boldsymbol{\theta}}_{\text{init}} - \boldsymbol{\theta}^*\right\|_1 \leq \frac{4\lambda_{\text{init}}s}{\phi^2(\boldsymbol{M}, s)}\right\}$. We have

$$\left\|\hat{\boldsymbol{\theta}}_{\text{init}} - \boldsymbol{\theta}^*\right\|_1 \geq \left\|(\hat{\boldsymbol{\theta}}_{\text{init}} - \boldsymbol{\theta}^*)_{S(\boldsymbol{\theta}^*)^c}\right\|_1 \tag{49}$$

$$= \sum_{j \in S(\boldsymbol{\theta}^*)^c}|(\hat{\boldsymbol{\theta}}_{\text{init}})_j| \tag{50}$$

$$\geq \sum_{j \in S(\hat{\boldsymbol{\theta}}_{\text{thres}}) \setminus S(\boldsymbol{\theta}^*)}|(\hat{\boldsymbol{\theta}}_{\text{init}})_j| \tag{51}$$

$$\geq |S(\hat{\boldsymbol{\theta}}_{\text{thres}}) \setminus S(\boldsymbol{\theta}^*)|\lambda_{\text{thres}}, \tag{52}$$

where (50) follows since $\boldsymbol{\theta}^*_{S(\boldsymbol{\theta}^*)^c} = 0$ by assumption, and (52) follows from the thresholding step in (5). Therefore, on the event $\mathcal{G}$, it holds that

$$|S(\hat{\boldsymbol{\theta}}_{\text{thres}}) \setminus S(\boldsymbol{\theta}^*)| \leq \frac{\left\|\hat{\boldsymbol{\theta}}_{\text{init}} - \boldsymbol{\theta}^*\right\|_1}{\lambda_{\text{thres}}} \leq \frac{4\lambda_{\text{init}}s}{\lambda_{\text{thres}}\,\phi^2(\boldsymbol{M}, s)}. \tag{53}$$

$\square$

For all $j \in S(\boldsymbol{\theta}^*)$, on the event $\mathcal{G}$, we have

$$|(\hat{\boldsymbol{\theta}}_{\text{init}})_j| \geq \theta_{\min} - \left\|(\boldsymbol{\theta}^* - \hat{\boldsymbol{\theta}}_{\text{init}})_{S(\boldsymbol{\theta}^*)}\right\|_\infty \tag{54}$$

$$\geq \theta_{\min} - \left\|(\boldsymbol{\theta}^* - \hat{\boldsymbol{\theta}}_{\text{init}})_{S(\boldsymbol{\theta}^*)}\right\|_1 \tag{55}$$

$$\geq \theta_{\min} - \left\|\boldsymbol{\theta}^* - \hat{\boldsymbol{\theta}}_{\text{init}}\right\|_1 \tag{56}$$

$$\geq \theta_{\min} - \frac{4\lambda_{\text{init}}s}{\phi^2(\boldsymbol{M}, s)}. \tag{57}$$

Therefore, if

$$\lambda_{\text{thres}} \geq \theta_{\min} - \frac{4\lambda_{\text{init}}s}{\phi^2(\boldsymbol{M}, s)}, \tag{58}$$

$S(\hat{\boldsymbol{\theta}}_{\text{thres}}) \supseteq S(\boldsymbol{\theta}^*)$ is satisfied on $\mathcal{G}$. Combining Lemma 1, (53), and (58) completes the proof of Theorem 1.

## D    Proof of Theorem 2

The proof of Theorem 2 closely follows the proof of Yang & Tan (2022, Th. 2). Therefore, we only explain the differences, which are as follows.

(i) Due to our construction, $m$ in Yang & Tan (2022) is replaced by $\tilde{T} = \left\lfloor \frac{T}{\lceil \log_2 d \rceil} \right\rfloor$.

(ii) Let $\mathcal{A}_r$ be the active arms in round $r$ and let $\{\boldsymbol{a}_r(i) : i \in \mathcal{A}_r\} \subset \mathbb{R}^{d_r}$ be the dimensionality-reduced arm vectors. From Fiez et al. (2019, Appendix B), it holds that

$$\max_{i \in \mathcal{A}_r} \|\boldsymbol{a}_r(i)\|^2_{\boldsymbol{M}(\tilde{\pi}^{(r)})^{-1}} \leq d_r \left(1 + \frac{d_r^2}{\tilde{T}}\right), \tag{59}$$

where $\tilde{\pi}^{(r)}$ is the rounded version of the G-optimal design output $\pi^{(r)}$.

(iii) In the proof of Yang & Tan (2022, Lemma 3), the set $\mathcal{B}_r$ is the set of arms in $\mathcal{A}_{r-1}$ excluding the best arm and $\lceil \frac{d}{2^{r+1}} \rceil - 1$ suboptimal arms with the largest mean rewards. We re-define $\mathcal{B}_r$ as the set of arms in $\mathcal{A}_{r-1}$ excluding the best arm and $\lceil \frac{d}{2^r} \rceil - 1$ suboptimal arms with the largest mean rewards.

With the modifications in items (i) and (ii) and following the steps in the proof of Yang & Tan (2022, Lemma 2), we get for any arm $i \in \mathcal{A}_{r-1}$

$$\mathbb{P}\left[\hat{\mu}_r(1) < \hat{\mu}_r(i) \mid 1 \in \mathcal{A}_{r-1}\right] \leq \exp\left\{-\frac{\tilde{T}\Delta_i^2}{8\lceil \frac{d}{2^{r-1}} \rceil (1 + \frac{d^2}{\tilde{T}})}\right\}, \tag{60}$$

where $\hat{\mu}_r(i)$ denotes the estimated mean of arm $i$ in round $r$.

Using item (iii) and (60), we go through the proof of Yang & Tan (2022, Lemma 3) and get

$$\mathbb{P}\left[1 \notin \mathcal{A}_r | 1 \in \mathcal{A}_{r-1}\right] \leq \begin{cases} \left(K - \frac{d}{2}\right) \exp\left\{-\frac{\tilde{T}\Delta_{\lceil\frac{d}{2^r}\rceil+1}}{16(\lceil\frac{d}{2^r}\rceil+1)(1+\frac{d^2}{\tilde{T}})}\right\}, & \text{if } r = 1 \\ \left(\frac{d}{2^r} + 1\right) \exp\left\{-\frac{\tilde{T}\Delta_{\lceil\frac{d}{2^r}\rceil+1}}{16(\lceil\frac{d}{2^r}\rceil+1)(1+\frac{d^2}{\tilde{T}})}\right\}, & \text{if } r > 1. \end{cases} \tag{61}$$

Finally, following the steps in the proof of Yang & Tan (2022, Th. 2) with (61), we get

$$\mathbb{P}\left[\hat{I} \neq 1\right] \leq \left(K - \frac{d}{2} + \sum_{r=2}^{\lceil\log_2 d\rceil} \frac{d}{2^r} + \lceil\log_2 d\rceil - 1\right) \exp\left\{-\frac{\tilde{T}}{16\left(1 + \frac{d^2}{\tilde{T}}\right)H_{2,\text{lin}}(d)}\right\} \tag{62}$$

$$\leq (K + \log_2 d) \exp\left\{-\frac{\tilde{T}}{16\left(1 + \frac{d^2}{\tilde{T}}\right)H_{2,\text{lin}}(d)}\right\}, \tag{63}$$

which completes the proof.

## E Proof of Corollary 1

We set $\kappa$, $\lambda_{\text{init}}$ and $\lambda_{\text{thres}}$ as

$$\kappa = \frac{16}{\phi^4\theta_{\min}^2}\frac{25}{24}, \tag{64}$$

$$\lambda_{\text{init}} = \frac{1}{\sqrt{\kappa(s + s^2)}}, \quad \text{and} \tag{65}$$

$$\lambda_{\text{thres}} = \frac{4}{\phi^2 s}\lambda_{\text{init}}. \tag{66}$$

Note that (66) sets $s_1 = s + s^2$ in (15), and for any $s \in \mathbb{N}$, we check that the condition in Theorem 3 holds:

$$\theta_{\min} = \frac{4}{\phi^2\sqrt{\kappa}}\sqrt{\frac{25}{24}} \geq \frac{4}{\phi^2\sqrt{\kappa}}\frac{s + \frac{1}{s}}{\sqrt{s + s^2}} = \lambda_{\text{init}}\left(c + \frac{4}{\phi^2}s\right). \tag{67}$$

Next, we would like to set $c_0 = \frac{T_1}{T_2}$ so that the two exponents in (14) are equal. However, since $H_{2,\text{lin}}(s + s^2)$ is not available to us, we use the lower bound

$$H_{2,\text{lin}}(s + s^2) = \max_{i \in \{2,\dots,s+s^2\}} \frac{i}{\Delta_i^2} \geq \frac{s + s^2}{\Delta_{s+s^2}^2} \geq \frac{s + s^2}{4}, \tag{68}$$

where the last inequality follows from the assumption $|\mu_k| \leq 1$ for all $k \in [K]$.[4] One can further upper bound $\Delta_{s+s^2}$ by using the values of $K$ arm vectors and searching for $\boldsymbol{\theta}^*$ that gives the largest $\Delta_{s+s^2}$.

We set the ratio $c_0 = \frac{T_1}{T_2}$ as

$$c_0 = \frac{25 \cdot 16 x_{\max}^2}{3\phi^4\theta_{\min}^2 \log_2(s + s^2)}, \tag{69}$$

which together with (64)–(68) ensures that

$$\exp\left\{-\frac{\left\lfloor\frac{T_2}{\log_2(s_1)}\right\rfloor}{16(1 + \epsilon)H_{2,\text{lin}}(s_1)}\right\} \geq \exp\left\{-\frac{T_1\lambda_{\text{init}}^2}{32 x_{\max}^2}\right\}. \tag{70}$$

Combining (70) with $s_1 = s + s^2$ completes the proof.

The ratio $c_0$ decreasing with $s$ as in (69) is consistent since as $s$ approaches $d$, the sparse linear bandit approaches the standard linear bandit, and we would expect to spend more budget on phase 2 than phase 1 for large $s$. We deliberately choose the parameters in (64)–(66) to maintain this property.

---

[4]This step is the only place where the assumption on the mean rewards is used.

# F   Implementation Details

Our code is available at https://github.com/recepcyavas/TMLR_sparse_linear_bandit. In all computations of the Lasso problem (4), we use the ADMM algorithm (Boyd et al., 2011).

## F.1   Lasso-OD with $K$-fold Cross-Validation

In the implementation of Lasso-OD-CV, the ratio of the budgets, $\frac{T_1}{T_2}$, is set to the default value $\frac{1}{4}$, i.e., naturally, the algorithm spends more budget for the BAI algorithm than for the support estimation.

For tuning the hyperparameters $\lambda_{\text{init}}$ and $\lambda_{\text{thres}}$, we pull $T_1$ arms according to the allocation given in (6). We use the following cross-validation steps to tune the parameters.

(i) Fix two sets of hyperparameters $\{\lambda_{\text{init},1}, \lambda_{\text{init},2}, \ldots, \lambda_{\text{init},m}\}$ and $\{\lambda_{\text{thres},1}, \lambda_{\text{thres},2}, \ldots, \lambda_{\text{thres},m}\}$ that are candidates for $\lambda_{\text{init}}$ and $\lambda_{\text{thres}}$ respectively.

(ii) Iteratively tune the parameters by fixing one of them and searching for the best parameter for the other one.

(iii) In each cross-validation round, the objective is to minimize the loss function

$$L = \frac{1}{T_1}\mathbb{E}\left[\left\|\mathbf{y} - \boldsymbol{X}\hat{\boldsymbol{\theta}}_{\text{thres}}\right\|_2^2\right] + c_1\mathbb{P}\left[\left\|\hat{\boldsymbol{\theta}}_{\text{thres}}\right\|_0 < s\right] + c_2\mathbb{E}\left[1\left\{\left\|\hat{\boldsymbol{\theta}}_{\text{thres}}\right\|_0 > s\right\}\left\|\hat{\boldsymbol{\theta}}_{\text{thres}}\right\|_0\right], \qquad (71)$$

where $c_1$ and $c_2$ are $\ell_0$-norm regularization parameters. Here, the first term in (71) is the mean-squared error; the second and the third terms penalize the $\ell_0$-norm error and force the hyperparameters to output an estimate with $s$ variables. In application, we set $c_1 = 200$ and $c_2 = 5$, giving more importance to detecting at least $s$ variables. If the value of $s$ is not available, we can still use this technique by setting $c_1 = c_2 = 0$. As standard, we approximate (71) by training the parameters in $K - 1$ blocks and testing in the remaining block.

(iv) To speed up convergence, in each round of cross-validation, we exponentially narrow down the candidate sets.

(v) To reduce the variance in cross-validation, we employ Monte-Carlo simulations, i.e., we independently partition the data into $K$ blocks for multiple times and then take the average loss.

## F.2   Lasso-OD-Analytical

In the implementation of Lasso-OD-Analytical, we set the hyperparameters $\lambda_{\text{init}}$ and $\lambda_{\text{thres}}$ as in (64)–(66), and $c_0$ is set so that (69) holds with equality. This setup effectively uses the hardness parameter $H_{2,\text{lin}}(s+s^2)$. To compute these hyperparameters, we first need to compute $\phi^2(\boldsymbol{M}, s)$. As discussed in Appendix A, this requires solving a MIDCP. We do this by using the YALMIP toolbox (Löfberg, 2004) because it does not ask to convert the problem into another one. Alternatively, one can use the CVX toolbox (Grant & Boyd, 2012) with a little bit more effort, or use Lemma 2 and solve an easier problem at the expense of some performance loss.

## F.3   OD-LinBAI and Other BAI Algorithms

We implement OD-LinBAI and other BAI algorithms shown in section 5 using the methods described in Yang & Tan (2022, Appendix E).

**BayesGap-Adaptive:**   In general, BayesGap algorithm (Hoffman et al., 2014) requires the knowledge of the hardness parameter. As in Hoffman et al. (2014); Yang & Tan (2022), at the beginning of each time instant, we input the estimated hardness parameter according to the three-sigma rule. In the experiments, we omit the oracle version of BayesGap that directly uses the knowledge of the hardness parameter.

## G  Additional Experiments

### G.1  Variants of Our Algorithm

We present two variants of Lasso-OD that modify the operations in phase 2 and one variant that replaces the thresholded Lasso in phase 1.

**Lasso-$\mathcal{X}\mathcal{Y}$-Allocation:**  This algorithm is identical to Lasso-OD except that the $G$-optimal design used to determine the allocations within each round is replaced by the $\mathcal{X}\mathcal{Y}$-allocation from Soare et al. (2014). Let $\mathcal{X} = \{a(i)\colon i \in \mathcal{A}\}$ be the set of arms in an active set $\mathcal{A}$. Let $\mathcal{Y} = \{\boldsymbol{x} - \boldsymbol{x}'\colon \boldsymbol{x}, \boldsymbol{x}' \in \mathcal{X}, \boldsymbol{x} \neq \boldsymbol{x}'\}$ be the set of arm differences. The $\mathcal{X}\mathcal{Y}$-allocation solves the problem

$$\pi^* = \arg\min_{\pi \in \mathcal{P}(\mathcal{A})} \max_{\boldsymbol{y} \in \mathcal{Y}} \|\boldsymbol{y}\|_{\mathsf{M}(\pi)^{-1}} . \tag{72}$$

Lasso-$\mathcal{X}\mathcal{Y}$-Allocation replaces Line 11 of Algorithm 4 with (72). In the experiments, we compute (72) using the Frank–Wolfe algorithm as in Fiez et al. (2019).

From Yang & Tan (2022, Proof of Lemma 2), the probability that a sub-optimal arm $i$ has a smaller estimated mean than the optimal arm 1 is bounded as

$$\mathbb{P}\left[\hat{\mu}(1) < \hat{\mu}(i)\right] \leq \exp\left\{-\frac{\Delta_i^2}{2\,\|\boldsymbol{a}(1) - \boldsymbol{a}(i)\|_{\boldsymbol{M}(\pi)^{-1}}}\right\} \tag{73}$$

$$\leq \exp\left\{-\frac{\Delta_i^2}{2\max_{i \neq j}\|\boldsymbol{a}(j) - \boldsymbol{a}(i)\|_{\boldsymbol{M}(\pi)^{-1}}}\right\} \tag{74}$$

$$\leq \exp\left\{-\frac{\Delta_i^2}{8\max_{i \in \mathcal{A}}\|\boldsymbol{a}(i)\|_{\boldsymbol{M}(\pi)^{-1}}}\right\}, \tag{75}$$

where $\pi$ is the allocation within the round. Here, (75) follows from the triangle inequality. The G-optimal design optimizes the allocation $\pi$ in (75), and $\mathcal{X}\mathcal{Y}$-allocation optimizes (74). Since $\mathcal{X}\mathcal{Y}$-allocation optimizes a tighter bound, Lasso-$\mathcal{X}\mathcal{Y}$-allocation is expected to perform better than Lasso-OD.

**Lasso-BayesGap:**  Since BayesGap performs better than OD-LinBAI in the examples in section 5, we propose the variant Lasso-BayesGap where in phase 2, OD-LinBAI is replaced by BayesGap-Adaptive from Hoffman et al. (2014).

In our implementations of Lasso-$\mathcal{X}\mathcal{Y}$-Allocation and Lasso-BayesGap (as described above), the parameters of Lasso are tuned via cross-validation.

**PopArt-OD:**  Recently, Jang et al. (2022) develop the PopArt algorithm, which estimates the unknown parameter $\boldsymbol{\theta}^*$ similarly to Lasso and TL. Due to its superior $\ell_1$ error to Lasso, PopArt also guarantees that the support of $\boldsymbol{\theta}^*$ is estimated efficiently. Similar to TL, the PopArt estimate is obtained by thresholding an initial estimate. Our variant, PopArt-OD, replaces the TL in phase 1 of Lasso-OD with PopArt, and retains phase 2 as is. We give its pseudo-code in Algorithm 5 and analyze its performance in the section below. Line 1 of PopArt involves an optimization problem that yields the optimal covariance matrix with respect to an upper bound on the error probability; this process reflects the design matrix optimization of TL described in Appendix A. In PopArt, if the population covariance $\boldsymbol{M}$ in Line 3 was replaced with the empirical covariance and if the Catoni estimator in Line 4 was replaced with averaging, the resulting algorithm would be the thresholded OLS estimator.

PopArt has one hyperparameter (the estimate on the error probability $\delta$ in Jang et al. (2022)). In the experiments to follow, we tune the hyperparameter using a $K$-fold cross validation procedure.

### G.2  Analysis of PopArt-OD

The following theorem bounds the error probability of PopArt-OD.

---

**Algorithm 5** PopArt-OD

---

**input** Time budget $T$, arm vectors $\boldsymbol{a}(1), \ldots, \boldsymbol{a}(K) \in \mathbb{R}^d$, $\theta_{\min}$, and sparsity $s$.

1: Solve the convex optimization problem $\nu^* = \arg\min\limits_{\nu \in \mathcal{P}[K]} \max\limits_{i \in [d]} \left( \{ \sum_{i=1}^K \nu_i \boldsymbol{a}(i) \boldsymbol{a}(i)^\top \}^{-1} \right)_{ii}$. Let the objective

value of the minimum be $H_*^2$ and $\boldsymbol{M} = \sum_{i=1}^K \nu_i^* \boldsymbol{a}(i) \boldsymbol{a}(i)^\top$. Set

$$\lambda_{\mathrm{PA}} = \min\left\{ \sqrt{2H_*^2}, \frac{\theta_{\min}}{2} \right\}, \quad c_{\mathrm{PA}} = \frac{2H_*^2}{\lambda_{\mathrm{PA}}^2 s \log_2 s}, \quad T_1 = \left\lceil T \frac{c_{\mathrm{PA}}}{1 + c_{\mathrm{PA}}} \right\rceil, \quad T_2 = T - T_1, \quad g = \frac{\lambda_{\mathrm{PA}}^2}{8H_*^2}. \tag{76}$$

2: Sample $T_1$ arms, $A_1, \ldots, A_{T_1}$ i.i.d. with $\nu^*$, and observe rewards $\mathrm{y}_1, \ldots, \mathrm{y}_{T_1}$.

3: For $t = 1, \ldots, T_1$, let $\tilde{\boldsymbol{\theta}}_t = \boldsymbol{M}^{-1} \boldsymbol{a}(A_t) \mathrm{y}_t \in \mathbb{R}^d$.

4: Set for $i \in [d]$, $\theta_i' = \mathrm{Catoni}\left( (\tilde{\boldsymbol{\theta}}_{1,i}, \ldots, \tilde{\boldsymbol{\theta}}_{T_1,i}), \sqrt{\frac{g}{(\boldsymbol{M}^{-1})_{ii}\left(1 + \frac{2g}{1-2g}\right)}} \right)$, where $\mathrm{Catoni}((Z_1, \ldots, Z_n), \alpha)$ is the

unique solution $y$ to the equation

$$\sum_{i=1}^n \psi(\alpha(Z_i - y)) = 0, \quad \psi(x) = \mathrm{sign}(x)(1 + |x| + x^2/2). \tag{77}$$

5: $\hat{\boldsymbol{\theta}}_{\mathrm{PA}} = \left( \theta_i' \mathbb{1}\{ |\theta_i'| \geq \sqrt{8\boldsymbol{M}_{ii}^{-1} g} \} : i \in [d] \right)$ and $\hat{S}_{\mathrm{PA}} = S(\hat{\boldsymbol{\theta}}_{\mathrm{PA}})$.

6: Run OD-LinBAI (Algorithm 4) restricted to the support $\hat{S}_{\mathrm{PA}}$ using $T_2$ pulls.

**output** $\hat{I}$ is the only remaining arm as the output of Algorithm 4.

---

**Theorem 4.** *For any linear bandit instance, the error probability of PopArt-OD given in Algorithm 5 is bounded as*

$$\mathbb{P}\left[ \hat{I} \neq 1 \right] \leq (K + \log_2 d + 2d) \exp\left\{ -\frac{T}{16\lfloor \log_2(s) \rfloor (1 + \epsilon_{\mathrm{PA}}) H_{2,\mathrm{lin}}(s)(1 + c_{\mathrm{PA}})} \right\}, \tag{78}$$

*where $c_{\mathrm{PA}}$ is defined in (76), below, and $\epsilon_{\mathrm{PA}} = \frac{(1 + c_{\mathrm{PA}})s^2}{T}$.*

*Proof.* By Line 5 of PopArt, if $i \in [d]$ satisfies that $\theta_i' \geq \lambda_{\mathrm{PA}}$, then $i \in \hat{S}_{\mathrm{PA}}$. From Jang et al. (2022, Prop. 1 and Th. 1), with probability at least $1 - 2d \exp\left\{ -\frac{T_1 \lambda_{\mathrm{PA}}^2}{8H_*^2} \right\}$, the initial PopArt estimator $\boldsymbol{\theta}'$ satisfies

$$\|\boldsymbol{\theta}' - \boldsymbol{\theta}^*\|_\infty \leq \lambda_{\mathrm{PA}}. \tag{79}$$

and the PopArt estimator satisfies $S(\hat{\boldsymbol{\theta}}_{\mathrm{PA}}) \subseteq S(\boldsymbol{\theta}^*)$. By selecting $\lambda_{\mathrm{PA}} \leq \frac{\theta_{\min}}{2}$, we further ensure that $S(\boldsymbol{\theta}^*) \subseteq S(\hat{\boldsymbol{\theta}}_{\mathrm{PA}})$, giving $S(\hat{\boldsymbol{\theta}}_{\mathrm{PA}}) = S(\boldsymbol{\theta}^*)$. Therefore, by the union bound and Theorem 2, the error probability of PopArt-OD is bounded as

$$\mathbb{P}\left[ \hat{I} \neq 1 \right] \leq \mathbb{P}\left[ S(\hat{\boldsymbol{\theta}}_{\mathrm{PA}}) \neq S(\boldsymbol{\theta}^*) \right] + \mathbb{P}\left[ \hat{I} \neq 1 \middle| S(\hat{\boldsymbol{\theta}}_{\mathrm{PA}}) = S(\boldsymbol{\theta}^*) \right] \tag{80}$$

$$\leq 2d \exp\left\{ -\frac{T_1 \lambda_{\mathrm{PA}}^2}{8H_*^2} \right\} + (K + \log_2 d) \exp\left\{ -\frac{T_2}{16\lceil \log_2 s \rceil H_{2,\mathrm{lin}}(s)(1 + \epsilon_{\mathrm{PA}})} \right\}. \tag{81}$$

The rest of the proof follows steps similar to the proof of Theorem 3, which aims to balance the two exponents in (81). The choices of $c_{\mathrm{PA}}, T_1$, and $T_2$ together with the lower bound $H_{2,\mathrm{lin}}(s) \geq \frac{s}{4}$ (see, (68)) imply that

$$\frac{T_1 \lambda_{\mathrm{PA}}^2}{8H_*^2} \geq \frac{T_2}{16\lceil \log_2 s \rceil H_{2,\mathrm{lin}}(s)(1 + \epsilon_{\mathrm{PA}})}, \tag{82}$$

which completes the proof. Note that we select $\lambda_{\mathrm{PA}} \leq \sqrt{2H_*^2}$ to ensure that $1 - 2g \geq \frac{1}{2} > 0$, making the Catoni parameter in Line 4 of Algorithm 5 valid. $\qquad \square$

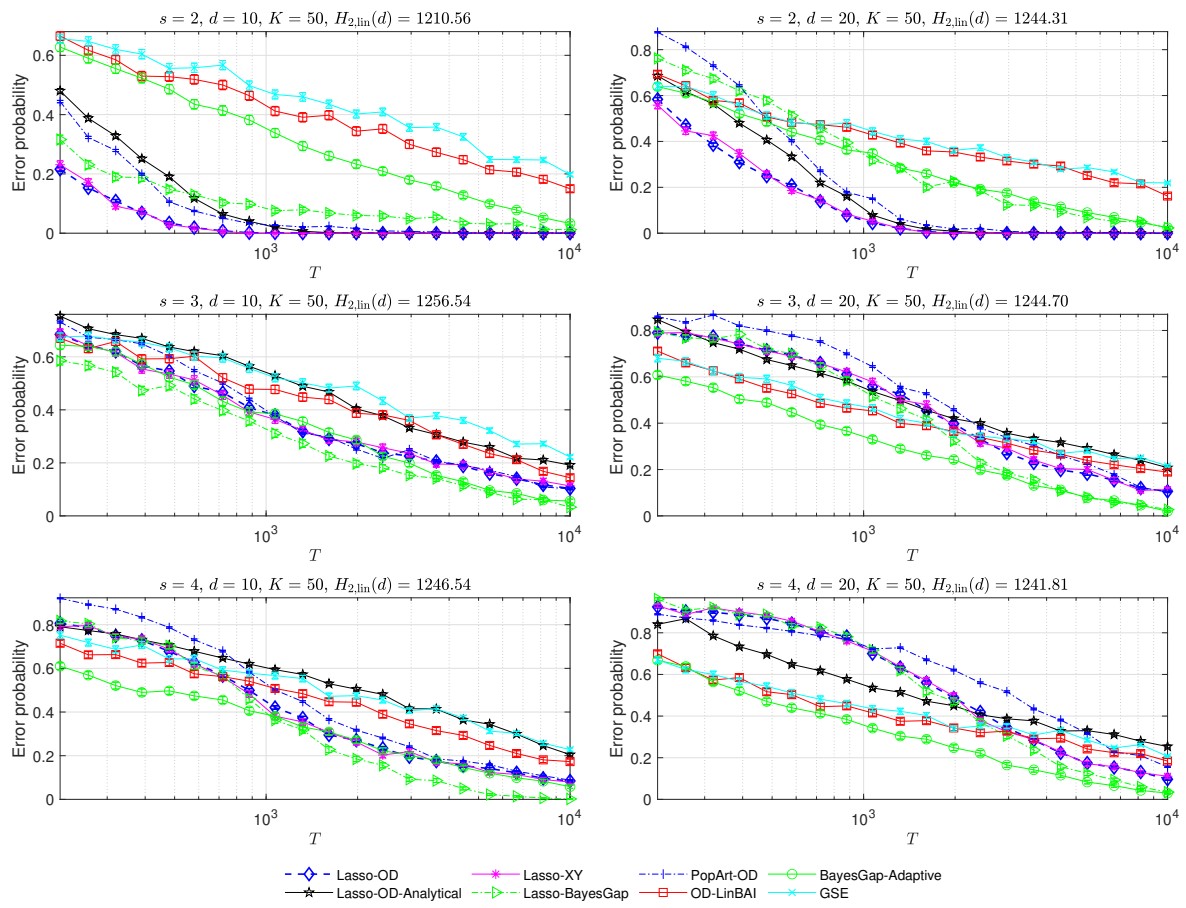

Figure 6: Comparison of several algorithms with $s \in \{2, 3, 4\}$.

Theorem 4 shows that PopArt-OD has an error probability that scales as

$$\exp\left\{-\Omega\left(\frac{T}{(\log_2 s) H_{2,\text{lin}}(s)}\right)\right\} \tag{83}$$

as $T$ and $s$ grow. This is the same theoretical result as that for Lasso-OD. However, we observe from the next section (specifically Appendix G.3.1) that Lasso-OD outperforms PopArt-OD empirically.

### G.3 Experiments

In the experiments below, we include Lasso-$\mathcal{X}\mathcal{Y}$-allocation and Lasso-BayesGap to the list of algorithms in Section 5.

### G.3.1 First Example

In the first example, we test the performance of the various BAI algorithms for sparsities of at least 2. We generate $K$ $d$-dimensional arm vectors $\boldsymbol{a}(k) = (\boldsymbol{a}(k)_i : i \in [d])$, $k \in [K]$, where $\boldsymbol{a}(k)_i$'s are distributed $\mathcal{N}(0, \frac{1}{s})$ independent across arms $k \in [K]$ and coordinates $i \in [d]$. The $s$-sparse unknown vector $\boldsymbol{\theta}^*$ is set as $\theta_i^* = \frac{1}{\sqrt{s}}$ for $i \in [s]$ and $\theta_i^* = 0$ for $i = s + 1, \ldots, d$. Figure 6 compares the performances of several algorithms in the literature and variants of our algorithm for $s \in \{2, 3, 4\}$, $K = 50$, $d \in \{10, 20\}$, and $T \in [200, 10^4]$. For all bandit instances under consideration, the performances of Lasso-OD and Lasso-$\mathcal{X}\mathcal{Y}$-allocation are almost identical. However, due to its low computational complexity (see Tables 3 and 4 for the CPU runtimes), Lasso-OD is preferred over Lasso-$\mathcal{X}\mathcal{Y}$-allocation. Among different variants of Lasso-OD, Lasso-OD and Lasso-$\mathcal{X}\mathcal{Y}$-allocation have the best performance for the instances with $s = 2$. For $s = 3$ and $s = 4$,

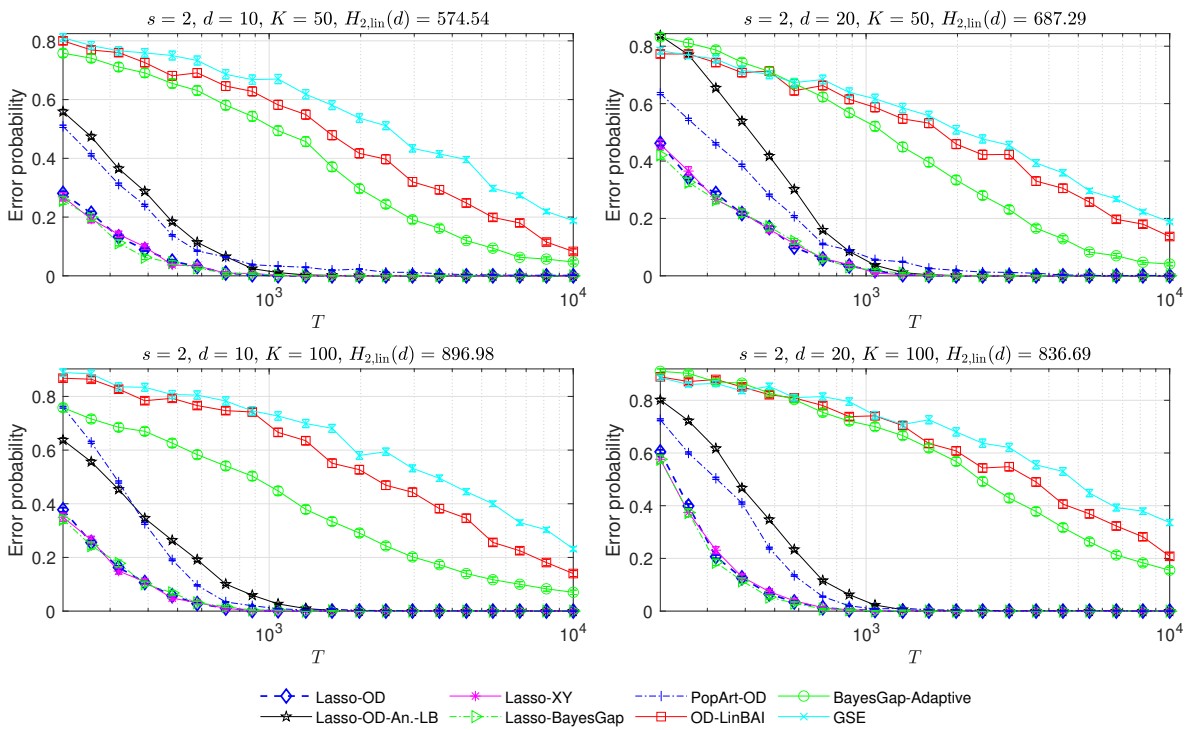

Figure 7: Comparison of several algorithms with $\boldsymbol{\theta}^*$ belonging to a finite set.

among the algorithms shown, Lasso-BayesGap performs the best for a large enough budget $T$. The poor performance of Lasso-based algorithms for small budgets is because at larger $s$, the minimum budget that should be allocated to phase 1 to reliably estimate the support of $\boldsymbol{\theta}^*$ increases with $s$. For the instances with $s \in \{3, 4\}$, BayesGap-Adaptive performs remarkably well, but it is outperformed by Lasso-based algorithms for $s = 2$. PopArt-OD algorithm is outperformed by Lasso-OD for all instances shown, which implies that the variable selection property of PopArt is poorer than that of the thresholded Lasso.

In Tables 3 and 4,[5] we report the average CPU runtimes for the instances in the first example with $s = 2$ and $s = 3$. Lasso-OD is superior to all other algorithms in terms of the computational complexity.[6]

### G.3.2 Second Example

In the second example, we assume that $\boldsymbol{\theta}^*$ belongs to the finite set $\mathcal{H} \triangleq \{\boldsymbol{\theta} \in \mathbb{R}^d \colon \|\boldsymbol{\theta}\|_0 = s, \theta_i \in \left\{-\frac{1}{\sqrt{s}}, 0, \frac{1}{\sqrt{s}}\right\}, \forall i \in [d]\}$. In other words, the non-zero coordinates of $\boldsymbol{\theta}^*$ are assumed to have magnitudes all equal to $\frac{1}{\sqrt{s}}$. We generate $K$ $d$-dimensional arm vectors in the vicinity of $\mathcal{H}$ as follows: $\boldsymbol{a}(k)_i = R_{k,i} \cos(\pi/4 + Z_{k,i})$, where $R_{k,i}$'s are independently and identically distributed (i.i.d.) generated with distribution $\text{Unif}(\{-1, 1\})$, $Z_{k,i}$'s are i.i.d. generated with distribution $\mathcal{N}(0, 0.01)$, and $R_{k,i}$'s and $Z_{k,i}$'s are independent. For this bandit instance, given the arm vectors and using the assumption that $\boldsymbol{\theta}^* \in \mathcal{H}$, we can lower bound the hardness parameter $H_{2,\text{lin}}(s + s^2)$ by computing the minimum hardness parameter for the vectors $\boldsymbol{\theta} \in \mathcal{H}$. In Figure 7, Lasso-OD-An.-LB computes the hyperparameters analytically and obtains $T_1$ from the lower bound on $H_{2,\text{lin}}(s + s^2)$ above instead of the true value of $H_{2,\text{lin}}(s + s^2)$. Figure 7 shows that Lasso-OD-An.-LB outperforms all other algorithms in the literature and achieves similar performance as Lasso-OD and Lasso-$\mathcal{XY}$-allocation for a large enough time budget.

---

[5]Pre-calculation in Tables 3 and 4 refers to the calculation of $\phi^2(\boldsymbol{M}, s)$ that is used to determine the hyperparameters for Lasso-OD-Analytical.

[6]All experiments are implemented on MATLAB 2023a on an Intel(R) Core(TM) i9-12900H processor.

Table 3: The empirical means of the CPU runtimes for $s = 2$, $d = 10$, $K = 50$.

| $T$ | CPU runtimes (milliseconds) | | | | | | | | |
|---|---|---|---|---|---|---|---|---|---|
| | Pre-calc. | Lasso Tuning | Lasso-OD | Lasso-$\mathcal{XY}$ | Lasso-BayesG. | Lasso-OD-An. | BayesGap-Ad. | OD-LinBAI | GSE |
| 100 | 9680 | 3300 | 0.98 | 9.0 | 1.6 | 1.7 | 3.2 | 2.2 | 4.0 |
| 200 | 9680 | 3500 | 0.61 | 5.2 | 2.9 | 1.2 | 5.1 | 2.2 | 4.0 |
| 400 | 9680 | 3800 | 0.55 | 3.6 | 5.5 | 0.83 | 9 | 2.1 | 4.0 |
| 800 | 9680 | 4380 | 0.48 | 3.4 | 11 | 0.76 | 17 | 2.1 | 4.0 |
| 1600 | 9680 | 6570 | 0.66 | 4.1 | 23 | 0.68 | 34 | 2.1 | 4.0 |
| 3200 | 9680 | 7520 | 0.66 | 4.0 | 46 | 0.70 | 69 | 2.1 | 4.0 |
| 6400 | 9680 | 7710 | 0.70 | 4.1 | 89 | 1.1 | 139 | 2.1 | 4.0 |

### G.3.3 Third Example

In the third example, we extend the example in Yang & Tan (2022); Jedra & Proutiere (2020); Fiez et al. (2019) to sparse linear bandits. We set $\boldsymbol{\theta}^* = (\frac{1}{\sqrt{2}}, \frac{1}{\sqrt{2}}, 0, \ldots, 0)$, i.e., $s = 2$, and $\mathcal{S} = S(\boldsymbol{\theta}^*) = \{1, 2\}$. For the coordinates in $\mathcal{S}$, we pull arms as in Yang & Tan (2022); we set $a(1)_{\mathcal{S}} = (\cos(\pi/4), \sin(\pi/4))$, $\boldsymbol{a}(K)_{\mathcal{S}} = (\cos(5\pi/4), \sin(5\pi/4))$, and $a(i)_{\mathcal{S}} = (\cos(\pi/2 + \phi_i), \sin(\pi/2 + \phi_i))$ for $i = 2, \ldots, K - 1$, where $\phi_i$ are independently drawn from $\mathcal{N}(0, 0.09)$. For any $i \in [K]$, we draw $a(i)_{\mathcal{S}^c}$ independently from the uniform distribution on the $(d - s)$-dimensional centered sphere of radius $\sqrt{\frac{d-s}{s}}$. Recall that since $\boldsymbol{\theta}^*_{\mathcal{S}^c} = 0$, the values of arms on the coordinates $\mathcal{S}^c$ have no effect on the best arm or the value of the hardness parameter. The problem would be identical to that in Yang & Tan (2022) if the agent knew the support $\mathcal{S}$. In this bandit instance, arm 1 is the best arm and there are $K - 2$ arms whose mean values are close to that of the second best arm. In the non-sparse case, i.e., $d = s = 2$, Yang & Tan (2022) demonstrate that OD-LinBAI outperforms the other algorithms. Figure 8 compares the performance of variants of our algorithm with the other algorithms in the literature. We report the empirical performances for $d \in \{10, 20\}$, $K \in \{50, 100\}$, and $T \in [200, 10^4]$. Among the algorithms shown, Lasso-OD and its variants Lasso-$\mathcal{XY}$-allocation and PopArt-OD significantly outperform the other algorithms. Unlike the previous two examples, for this example, Lasso-BayesGap is not the best performing algorithm.

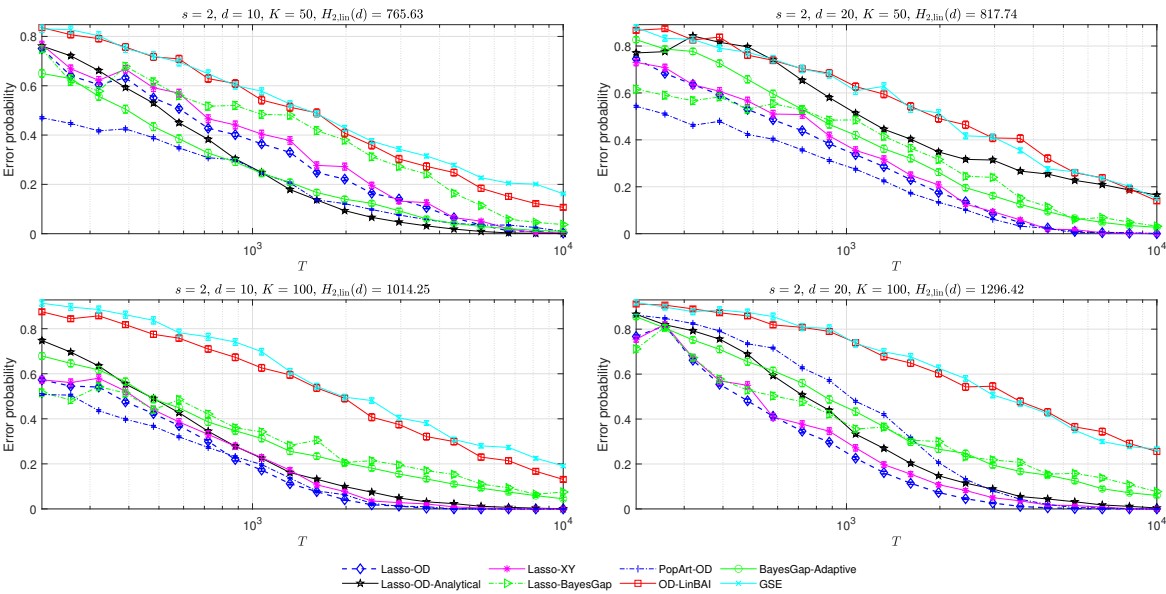

Figure 8: Comparison of several algorithms for the example bandit instance in Yang & Tan (2022).

Table 4: The empirical means of the CPU runtimes for $s = 3$, $d = 10$, $K = 50$.

| $T$ | CPU runtimes (milliseconds) | | | | | | | | |
|---|---|---|---|---|---|---|---|---|---|
| | Pre-calc. | Lasso Tuning | Lasso-OD | Lasso-$\mathcal{X}\mathcal{Y}$ | Lasso-BayesG. | Lasso-OD-An. | BayesGap-Ad. | OD-LinBAI | GSE |
| 100 | 27100 | 4200 | 1.1 | 12 | 1.6 | 2.2 | 2.5 | 2.4 | 4.2 |
| 200 | 27100 | 4500 | 0.96 | 8.9 | 2.9 | 2.1 | 5.2 | 2.4 | 4.1 |
| 400 | 27100 | 4900 | 0.94 | 8.1 | 5.7 | 2.3 | 10 | 2.4 | 4.1 |
| 800 | 27100 | 5000 | 0.98 | 7.6 | 12 | 2.3 | 17 | 2.4 | 4.2 |
| 1600 | 27100 | 5100 | 1.5 | 9.2 | 24 | 2.1 | 34 | 2.4 | 4.2 |
| 3200 | 27100 | 7000 | 1.5 | 9.2 | 47 | 2 | 68 | 2.4 | 4.2 |
| 6400 | 27100 | 9800 | 1.6 | 9.4 | 93 | 2 | 137 | 2.4 | 4.1 |

### G.3.4 Fourth Example

In the final example, we test the performance of thresholded Lasso in which the whole horizon of length $T$ is used for learning the support of $\boldsymbol{\theta}^*$. We draw each entry of the design matrix $\boldsymbol{X} \in \mathbb{R}^{T \times d}$ i.i.d. from $\mathcal{N}(0, \frac{1}{s})$ and set $\boldsymbol{\theta}^* = (\frac{1}{\sqrt{s}}, \ldots, \frac{1}{\sqrt{s}}, 0, \ldots, 0)$ where $\boldsymbol{\theta}^*$ has $s$ non-zero entries. In Figure 9, we report the empirical probability of detection error $\mathbb{P}\left[S(\hat{\boldsymbol{\theta}}_{\text{thres}}) \not\supseteq S(\boldsymbol{\theta}^*)\right]$ and the empirical mean $\mathbb{E}[|S(\hat{\boldsymbol{\theta}}_{\text{thres}})|]$ over 10,000 independent trials. For $s = 2$, the empirical error probability is 0 for $T \geq 400$; for $s = 4$, the empirical error probability is 0 for $T \geq 800$. Figure 9 shows that for $T \geq 100$ and $s \in \{2, 4\}$, thresholded Lasso is capable of correctly detecting the active variables in $\boldsymbol{\theta}^*$ with high probability while also keeping the average number of false positives close to zero. As expected, the average number of false positives increases with $s$.

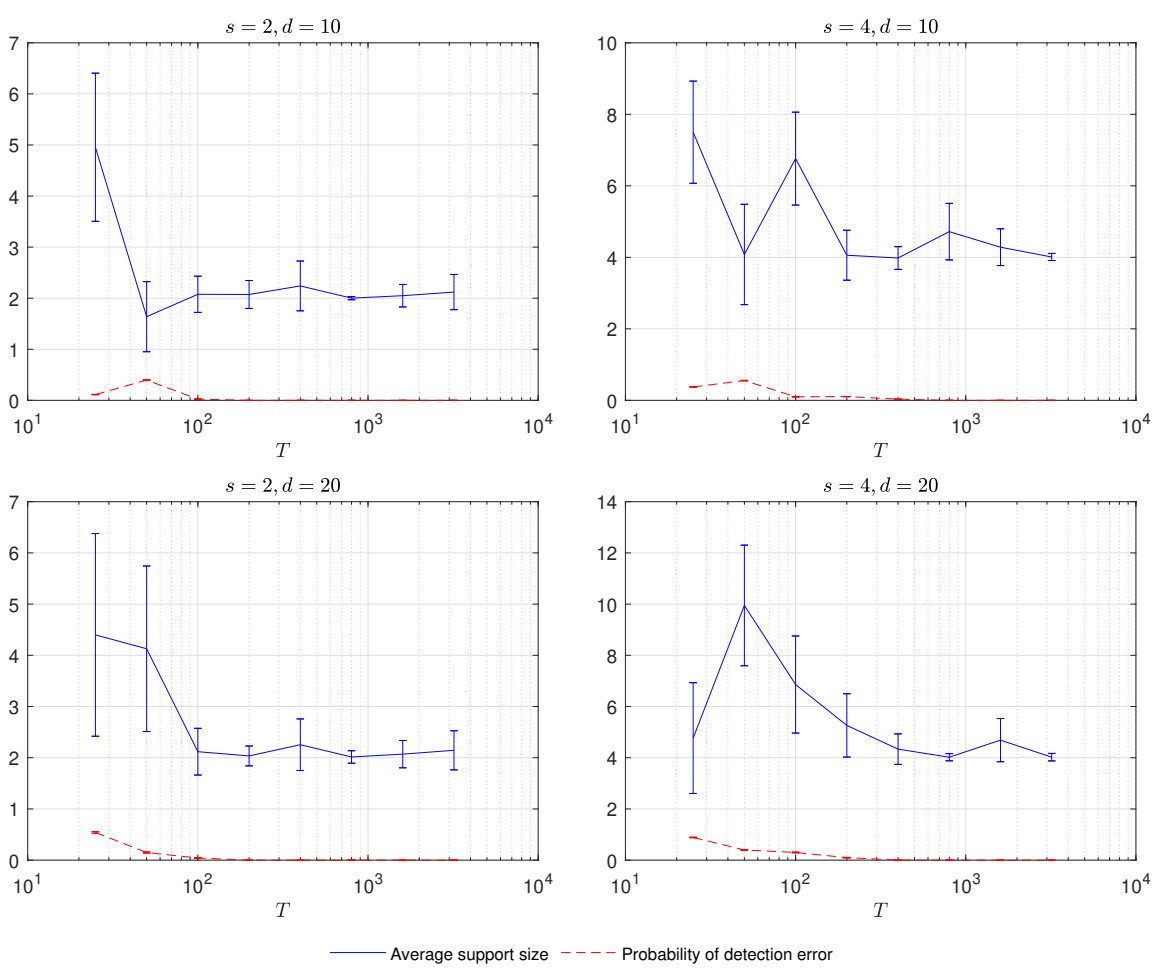

Figure 9: The empirical detection error probability and the empirical size of the thresholded Lasso output.

