# OpenReview forum: "Fixed-Budget Best-Arm Identification in Sparse Linear Bandits"
_TMLR — Accepted by TMLR_

### Review · Reviewer_EjKY · 2023-11-13

**Summary Of Contributions:**

The paper considers the problem of finding the arm with largest reward mean in a linear bandits environment given a fixed budget of samples. The paper also bounds the error probability of their algorithm showing it depends on the sparsity parameter. Experimental results show the proposed algorithm outperforms the existing bechmarks.

**Audience:**

Yes

**Broader Impact Concerns:**

Not needed.

**Claims And Evidence:**

Yes

**Requested Changes:**

I suggest including few dataset experiments and comparing the algorithm to sparse algorithms by replacing OD-LinBAI with other FB-BAI algorithms.

**Strengths And Weaknesses:**

Strengths:
1. The paper aligns well with the existing work around fixed BAI algorithm and covers the literature properly.
2. The error bounds are compared to similar algorithms and proven to be SOTA.
3. Extensive experimental results demonstrate the behavior of the algorithm properly.

Weaknesses:
1. The paper does not include real-world dataset experiments to study the robustness of the algorithm w.r.t its assumptions.
2. A simple benchmark that could be used is to replace the OD-LinBAI algorithm with other FB-BAI algorithms. This is also missing.

---

> ### Author Response · Authors · 2023-12-18
>
> We thank the reviewer for their time, acknowledging our contribution, and making constructive suggestions, which we implemented in the revised paper.
>
> 1. We have now added a real-world dataset experiment where the dimension $d = 55$, $s = 3$, and $K = 500$. Similar to the synthetic data experiments, we observed that the proposed Lasso-OD algorithm has the smallest error probability among the tested algorithms.
>
> 2. In Appendix G, we had already experimented with different settings where OD-LinBAI is replaced with some other state-of-the-other algorithms such as BayesGapAdaptive and $\mathcal{X}\mathcal{Y}$-Allocation. BayesGapAdaptive is a similar performance as the OD-LinBAI in some cases; however, its computational complexity is much larger. Therefore, OD-LinBAI (as the algorithm to use in the second phase) is still preferred.

---

### Review · Reviewer_JhRB · 2023-11-18

**Summary Of Contributions:**

This paper deals with the best-arm identification problem in sparse linear bandits under the fixed-budget setting.

Authors designed a two-phase algorithm, Lasso and Optimal-Design- (Lasso-OD) based linear best-arm identification.

1) The first phase of Lasso-OD leverages the sparsity of the feature vector by applying the thresholded Lasso [Zhou et al 2009], which estimates the support of the hidden parameter vector with high probability using rewards from the appropriate experimental design.
2) The second phase of Lasso-OD applies the OD-LinBAI algorithm by Yang and Tan (2022) on that estimated support.

For fixed sparsity s and budget T, the exponent in the error probability of Lasso-OD depends on s but not on the dimension d which is noticeable.

Furthermore, the authors show a lower bound for this problem and tried to express minimax optimality in their result.

Finally, the authors provide numerical examples to demonstrate the significant performance improvement over the existing algorithms for non-sparse linear bandits.

**Audience:**

Yes

**Broader Impact Concerns:**

There are no concerns about ethical implications.

**Claims And Evidence:**

Yes

**Requested Changes:**

1) (Minor) I hope authors justify why they should rely on the E-optimality instead of $x_{\max}$. If they could optimize $x_{\max}$ easily, it will be better for their theoretical performance. \\\\


2) (Minor) I hope the authors state the scale difference between the 'minimum eigenvalue', 'compatibility constant' they approximated, '$x_{\max}$, and 'H*' in the PopArt paper. According to my understanding, these are the quantities that depend on the geometry of the action set and determine the error of the regression. Especially, it would be great if they could find an example that $x_{\max}$ shows a big difference with other action set geometry constants (like minimum eigenvalue).\\\\

3) (Minor) I hope authors include 'minimum signal' related arguments. What happens when the agent doesn't have the minimum signal assumption? Please let me know if I missed this argument. \\\\

4) (Minor) How far is the 'relaxed optimization' of MIDCP in Appendix A versus the true compatibility constant? What about the gap between the result of QP after relaxing the $l_1$ norm condition versus the true compatibility constant?\\\\

5) (Minor) What happens if we apply Thresholded lasso in a sparse linear fixed-confidence BAI problem and use the existing fixed-confidence BAI algorithm, as you did in a fixed-budget setting? Would it easily derive a new sparse-linear fixed-confidence BAI? What is the major difference or difficulty in this approach?

**Strengths And Weaknesses:**

Overall, I think this is an interesting paper with enough interesting analysis on sparse linear regression.



Strength:
1) It is the first algorithm specialized for the sparse linear BAI.
2) The idea is straightforward - identify support using Lasso, thresholding out the unnecessary coordinates, and apply the known linear BAI algorithm.
3) Since the idea itself is simple, seems like there's no specific place to make a wrong proof.
4) They did a thorough research on the sparse linear estimation. It helped me to understand a bit more about sparse linear regression.



Weakness:

1) They directly used the compatibility constant in their result. Usually, this constant is very difficult to optimize. They even mentioned in Appendix A that it is computationally intractable. Therefore, even though they used 'compatibility constant' in their analysis, actually they used 'E-optimality' (maximizing the minimum eigenvalue) and approximated the compatibility constant by the minimum eigenvalue (Lemma 2 in Appendix A).

1-1) More interestingly, they eventually made their final error probability bound based on $x_{\max}$ (Theorem 3), which is a kind of 'maximum diagonal entry of the covariance matrix.' I am wondering, if their theoretical result depends on this '$x_{\max}', why don't they optimize this quantity? Why the authors should rely on the E-optimal experimental design and compatibility constant argument?


2) Their problem relies on the assumption that they know the minimum signal, $\theta_{\min}$. It usually makes the problem much easier.

---

> ### Author Response · Authors · 2023-12-18
>
> We thank the reviewer for their careful reading and making constructive suggestions that will further improve our paper.
>
> Weaknesses
>
> 1. We use both the compatibility constant and the minimum eigenvalue of the covariance matrix in our design. The computation of the compatibility constant given a fixed design matrix $X$ is already difficult. Optimization (finding the optimal arm allocation (equivalently, $X$) that maximizes the compatibility constant) over it is even more difficult since the problem becomes no longer convex. Yet, in the high dimensional scenario, we need to optimize the minimum compatibility constant over $\approx d^s$ subsets of $[d]$ of size $s$, each being lower bounded by the minimum eigenvalue. Since $d$ is assumed to be large, the minimum compatibility constant over $d^s$ subsets is close to the minimum eigenvalue for any arm allocation. Therefore, for large $d$, optimization over the minimum eigenvalue is a feasible and sufficiently accurate approach. We use the compatibility constant to get a refined bound on the error probability related to the Lasso phase, which is then used to optimize the $T_1/T_2$ ratio.
> 2. Yes, our algorithm assumes the knowledge of minimum signal strength $\theta_{\min}$, which is admittedly a weakness. It is possible to derive an alternative to Theorem 1 that controls the performance of thresholded Lasso without relying on a minimum-signal strength assumption. To do this, we should replace the bound on van de Geer and Buhlmann's result in [Buhlmann and van de Geer, Theorem 7.8], where they show
> $$ |S(\hat{\theta}) \setminus S(\theta^*)| = \frac{1}{\phi^4_{\min}(6, S(\theta^*), 2s)} \frac{\lambda_{\mathrm{init}}^2}{\lambda_{\mathrm{thres}}^2} O(s).$$
> Here $\phi^4_{\min}(6, S(\theta^*), 2s)$ is a compatibility-constant-like constant. Unfortunately, they do not derive the coefficient of $O(s)$ explicitly; and it seems difficult to derive an explicit form for the $O(s)$ term from their proof technique. Therefore, we preferred to use the version with the minimum signal assumption. If the compatibility constants ($\phi$ and $\phi_{\min}$) are lower bounded by a constant independent of $s$, both versions yield $O(s)$ false positives.
>
> Requested changes:
> 1. As you mentioned, ideally, one should include the $x_{\max}$ in the design matrix optimization. However, in practice, we usually normalize the data so that the squared norm over all arms are equal for each attribute $j \in [d]$. It should be reasonable to assume that $x_{\max}(\nu)$ does not vary too much with respect to the allocation vector $\nu$. In addition, if we seek to include $x_\max$ in our optimization of $\nu$, the problem no longer satisfies the MIDCP constraints; therefore, we can no longer solve it efficiently. We have added a new discussion about this issue in Appendix A.
> 2. You are correct that the quantities $\sigma_{\min}$, $\phi^2(M, s)$, and $H_*^2$ of Jang et al. depend on the geometry of the instance. Jang et al. 2023 established the relationship $\frac{1}{d H_*^2} \leq \sigma_{\min} \leq \frac{1}{H_*^2}$. However, the case $\sigma_\min \approx \frac{1}{d H_*^2}$ occurs iff all the arms are close to each other in $\ell_2$ distance, i.e., $a(i) - a(1) = o(1) \boldsymbol{1}$ for all $i \in [K]$, which is not a reasonable assumption. In most instances of interest, $\sigma_{\min} \approx \frac{1}{H_*^2}$ holds. Secondly, the compatibility condition is much weaker than the minimum eigenvalue condition. There are instances where $\sigma_{\min} \approx \frac{1}{H_*^2} \to 0$ but $\phi^2(M, s)$ is a positive constant.
> If we have a dataset whose attributes are not balanced, the optimization over $x_{\max}$ can yield a better allocation vector $\nu$. However, as mentioned in the previous item, that refined optimization problem is unfortunately intractable.
> 3. Please see our response in Weakness section item 2.
> 4. The only relaxation we do in the computation of the compatibility constant is to allow fractional arm pulls, which creates a multiplicative factor of $1 + O(1/T_1)$ compared to the true value of the compatibility constant. We did not need to use the QP relaxation to compute the compatibility constant for a fixed design matrix. The challenge is to optimize the compatibility constant with respect to the feasible design matrices. Note that this optimization with the QP relaxation of the compatibility constant is not a QP, therefore is not tractable.
> 5. Our first idea was to use our approach in both fixed-budget and fixed-confidence settings. We realized that the performance of the state-of-the-art algorithms for the fixed-confidence setting (e.g., Tao et al. 2018 and Fiez et al. 2019) is not heavily dependent on the dimension $d$. Therefore, although we can apply our approach to the FC setting in a similar way, the improvement on the error probability compared to the state-of-the-art algorithms is not very significant. Because of this reason, we did not choose to do that in this paper.

---

> ### Comment · Reviewer_JhRB · 2023-12-21
> **Questions about the rebuttal**
>
> Thanks for your detailed rebuttal, and all the changes you made. It was really helpful to understand. I have a few minor questions regarding your answers.
>
> Regarding Change 2)
>
> You mentioned $\sigma_{min} \approx \frac{1}{dH^2}$ occurs iff all the arms are close to each other in $\ell_2$ distance. As far as I know, in Jang et al they mentioned only one example. Do you have a proof about it? I'm interested in that.
>
> Regarding Change 1)
>
> To be honest, I am not sure whether normalizing $\sum a(k)^2_j$ is usual... I know many people assume the boundedness of $||a(k)||$ for various types of norms (like $l_2$ and $l_\inf$), but I haven't seen normalizing $k$-th coordinate specifically for all arms...

---

> > ### Author Response · Authors · 2023-12-22
> > **Response to Reviewer's Comments**
> >
> > Thank you for your quick reply.
> >
> > 1. Unfortunately, we do not have a proof for such a strong statement. We should have written our claim more precisely as follows.
> >
> > In the example in Appendix D.2 of Jang et al.'s PopArt paper, the authors show that $\sigma_{\min} \approx \frac{1}{d H_*^2}$ for a design matrix that satisfies $a(k_1) - a(k_2) = o(1) \mathbf{1}$ for all $k_1, k_2 \neq 1$.
> >
> > So, we do not know whether the "reverse implication" holds, i.e., whether or not $\sigma_{\min} \approx \frac{1}{d H_*^2}$ is true if $a(k_1) - a(k_2) = o(1) \mathbf{1}$ is **not** satisfied for some $k_1,k_2\ne 1$.
> >
> > 2. On pages 107-108 of Buhlmann and van de Geer's book, the authors argue that in practice, $\hat{\sigma}j^2 = \frac{1}{K} \sum_{k = 1}^K a(k)_j^2$ for all coordinates $j$ can be normalized to be the same, e.g., to 1. This roughly says that all coordinates have the "same importance". If $\hat{\sigma}_j^2$'s are not of the same order across all $j$ (e.g., some features have units in meters while others have units in millimeters), Lasso will prioritize the coordinates with the large $\hat{\sigma}_j^2$ and the role of the other coordinates becomes negligible, which is undesirable.
> >
> > Note that the normalization step is equivalent to defining the Lasso as
> >
> > $$ \hat{\theta} = arg \min_{\theta} \frac{1}{n} || Y - X \theta ||_2^2 + \lambda \sum_j \hat{\sigma}_j |\theta_j|,$$
> >
> > which introduces weights to the Lasso parameter, or that the $\ell_1$ norm is weighted by the $\hat{\sigma}_j$'s.

---

> > > ### Comment · Reviewer_JhRB · 2024-01-01
> > > **Last minute request**
> > >
> > > Sorry for the last-minute comment, but I strongly want to suggest adding 'action set geometry constant' for your order comparisons.
> > >
> > > It made me misunderstand your result, and spent some time checking the order of the 'action set geometry constant.'
> > >
> > > 1) On page 9, you mentioned such as like this
> > >
> > > > The error probability exponent is independent of the dimension $d$
> > >
> > > However, I believe in many cases, the action set geometry constants such as $x_{max}$, $C_{min}$, or even $H^*$ should depend on $d$ when the action set is bounded (which is a traditional assumption in linear bandit type problems, though the norm itself could be varied - like $l_2$ or $l_\infty$). For example, when the action set is a Frobenius norm ball, I believe you can't find your $b$ and $x_{max}$ to be dimension-independent. This sentence is very misleading.
> > >
> > > 2) Not only that, in many of the authors' statements, the authors omitted the 'action set geometry constant', and only stated $s$ and $T$  for the order comparison. I hope the authors add action set geometry constant for all order notations.
> > >
> > > Especially, between Eq. (19) and Eq. (20) on page 8, you mentioned two misleading statements:
> > >
> > > > Misleading1) The ratio c0 scales as $\Theta(\frac{1}{\phi^4 \log (s)})$
> > >
> > > > Misleading2) Since $c_0$ in (19) is lower bounded by a positive constant for all $s \in N$,
> > >
> > >
> > > For Misleading1, your $c0$ scales as $\Theta(\frac{x_{\max}^2}{\phi^4 \log (s)})$. This is important for the following example:
> > >
> > > 1. Imagine when the action set is a set of canonical vectors: $\mathcal{A}=\{e_1, e_2, \cdots, e_d\}$ and fix $\theta^*$. Then you will have some probability error upper bound $p$.
> > > 2. Now imagine that you shrink your action set by small constant $\xi<<1$ as $\xi \mathcal{A}=\{\xi e_1, \xi e_2, \cdots, \xi e_d\}$. Let's say the error probability in this case is $p_{\xi}$
> > > 3. Then according to your Misleading1, you are saying $p_{\xi} = p / \xi^4$ which is a terrible ratio.
> > > 4. Actually your ratio is $x_{max}^2 / \phi^4$ which means $p_{\xi} = p/\xi^2$. Now that you have the same growth rate w.r.t $\xi$ as PopArt-OD.
> > >
> > > This survey is caused by the absence and wrong description of the action-set geometry order in your short results.
> > >
> > > All the intermediate constants, like $b$ (you can use just compatibility constant and not mention $b$ at all) confuse me. Even I first thought your $b$ is some independent constant from the action set and sampling distribution, and I had to read all the details of the proofs in the Appendix to check whether I should reject your paper (that's why I asked 'Why didn't you use $x_{max}$ optimized sampling distribution?' question in the first comment). This is a clarity problem.
> > >
> > > For Misleading2, I failed to find any assumption about the scale of the compatibility constant. The only I can find is the assumption that $<\theta^*, A> \leq 1$, and this is true for the shrinking example above for all $\xi$. Your compatibility constant can grow infinitely in our current setting.
> > >
> > >
> > >
> > > As I start mentioning the action set geometry comparison, do you have any idea about the scale difference between $H^2_*$ of PopArt-OD and $x^2_{max}/\pi^4$ of Lasso-OD? It would be great if you elaborate on that more.

---

> > > > ### Author Response · Authors · 2024-01-02
> > > > **Our response to "last minute request"**
> > > >
> > > > We thank the reviewer for raising up an important issue in the paper. We updated the paper to clarify the asymptotic interpretation of our main result. The updated section is in blue after Corollary 1.
> > > >
> > > > As the reviewer mentioned, the constants such as $C_{\min}$ (minimum eigenvalue), $\phi^2$, $x_{\max}^2$, and $H_*^2$ depend on the geometry of the arm vectors (or, the action set). One can construct instances, where these constants may or may not depend on the dimension $d$. For example, in the first example you gave (the standard basis vectors), $\phi^2 = C_{\min} = 1$, independent of $d$. Therefore, as you mentioned, when we talk about asymptotic scalings, we should clarify how these parameters scale with the quantity or quantities approaching infinity.
> > > >
> > > > In the statements of the previous version, we implicitly assumed that $x_{\max}^2$ is upper bounded by some $a_0^2$ independent of $s$ and $d$, and $\phi^2$ is lower bounded by some $\phi_0^2$ independent of $s$ and $d$. Note that $x_{\max}^2 \leq \max_{k \in [K]} || a(k)||_{\infty}$, and it is reasonable to assume that the maximum coordinate of any arm is bounded. The $\phi^2 \geq \phi_0^2$ assumption is justified by orthonormal arm vectors as mentioned above. Then, with these assumptions, $c_0$ scales as $\Theta(\frac{1}{\log s})$, which approaches 0 (assuming $s\to\infty$) or $O(1)$ (assuming $s$ is fixed), both yielding the desired scaling stated after Corollary 1. We would also like to remind that our main results (Theorems 2-3 and Corollary 1) are all non-asymptotic bounds; therefore, we do not need to introduce the scaling assumption for the main theorems. We rewrote the paragraph around eq. (20) by explicitly stating the asymptotic assumptions we make. The issues are corrected as: Misleading 1--> $c_0$ scales as $\Theta(\frac{1}{\log s})$, Misleading 2--> since $c_0$ is $O(1)$ (For our conclusion, a bounded $c_0$ is sufficient.)
> > > >
> > > > The second example: If arms are scaled by $\xi \to 0$ as $d$ grows, then the Gram matrix approaches the zero matrix, and $\phi^2$ tends to 0. Therefore, this example does not satisfy the required conditions. In this scenario, the new $c_0$ would be multiplied by $\frac{1}{\xi^2}$, so we spend more time in phase 1, i.e., $T_1$  is larger. (This $c_0$ calculation uses the assumption $|\mu_k| \leq 1$. In this setting this upper bound becomes loose, which causes the algorithm to allocate more time/budget on Lasso.) As a result, $H_{2, lin}(s + s^2)$ also grows since the arms get closer, and the error probability increases. This is intuitive since in this setting, the distributions of the observations become closer to that of pure noise. Thus, detecting the support of $\theta^*$ becomes increasingly difficult. Naturally, we should not expect low error probabilities for such hard instances.
> > > >
> > > > Currently, we do not have bounds on the relationship between $1/\phi^2$ and $H_*^2$. But, for example, when the arm vectors are the standard basis vectors $e_i \in \mathbb{R}^d, i \in [d]$, both quantities are 1, independent of $d$. The constraints in the definition of the compatibility constant makes it cumbersome to relate it, in a simple way, to $H_*^2$.
> > > >
> > > > Since the $b = \frac{4}{\phi^2}$ notation is potentially confusing, we have removed it. Thanks for the suggestion.

---

### Review · Reviewer_oj4o · 2023-12-04

**Summary Of Contributions:**

In this study, they study a fixed budget setting for best arm identification under sparse structure. Given a budget T, the learner aims to find the best arm while minimizing the error probability. The paper considers the sparsity and the goal is to develop an algorithm whose error probability only depends on the sparsity parameter s:=\|\theta^*\|_1 rather than the original dimension d. They also conduct an experimental evaluation on a small dataset.

**Audience:**

Yes

**Claims And Evidence:**

Yes

**Requested Changes:**

1. Experiments for the case when d is sufficiently large.

2. Hope that the authors will address my concern written in Weakness above.

3. Could you provide an explanation on how to compute T_1/T_2 and decide parameters lambda in the main text? What is the order of b (equivalently, what the order will compatibility constant \psi(M,s) become when we use such an optimal parameter choice in the end)?

**Strengths And Weaknesses:**

Strengths:

1.This study first addresses the sparse linear bandit for BAI in the fixed budget setting.

2. The proposed algorithm is computationally efficient and enjoys the desirable upper bound where error probability is independent of the original dimension d. This property is especially useful in high-dimensional settings.

3. The paper is easy to follow, and all the related work is clearly discussed.


Weakness:
1. Theoretical results are not significant, although novel results. Indeed, the proposed algorithm is a naive combination of Lasso and the existing linear bandit algorithm. In particular, the first phase is an adaption of E-optimal design [Hao+2020] to determine the optimal allocation of arm strategy and [Ariu+2020] to find support including the true support with high probability. The second phase is to run OD-LinBAI [Yang and Tan 2022]. The important analysis highly relies on [Ariu+2020] for the concentration bound that relates sparsity parameter s and compatibility constant \psi(M,s). Combining them to derive the bound for fixed budget setting is not a significant effort.

2. The experimental results are not satisfying. Although the motivation of the problem is from the high-dimensional cases, the experiments are only conducted on a very small dataset. It would be more reasonable to show the high-dimensional instances such as d=1000 or 10000 as in [Ariu+2020].

3.  The techniques discussed here seems also applicable to fixed confidence settings. Although the authors already mentioned the FC setting in the Conclusion, I would not understand why it is not worth studying FC setting. If there is no result in the sparse linear bandit for the FC setting, it is still interesting to investigate FC setting. As a naive solution for FC setting, we can still use E-optimal design until we can identify \hat{S} including the true support with a high probability, and after that, we can conduct a phased elimination algorithm based on G-optimal design like [Tao+2018] for FC setting instead of OD-LinBAI [Yang and Tan 2022]. I believe this algorithm can solve FC setting and have a sample complexity bound which is independent of the original dimension d.

---

> ### Author Response · Authors · 2023-12-18
>
> We thank the reviewer for their detailed reading and making several constructive suggestions, which we implemented in the revised manuscript.
>
> Weaknesses:
>
> 1. We believe that choosing the parameters $T_1$, $\lambda_{init}$ and $\lambda_{thres}$ judiciously to "balance" the Lasso and BAI error terms and thus to show that the resulting error probability exponent is independent of $d$ (but dependent on the sparsity $s$) is rather non-trivial, and is the key contribution of our work. It is true that our algorithm combines several tools that appeared in other papers in the literature, but our problem and our results are different than those. A naive combination of the existing results from Lasso and FB BAI will not be minimax optimal in the sense of attaining the lower bound; it is through choosing the parameters of the length of the first phase $T_1$ and regularization/threshold parameters such as $\lambda_{thres}$ that allow us to obtain a neat and optimal (in the exponent) expression for the upper bound on the failure probability of finding the best arm.
>
> 2. The main reason why we chose relatively small $d$ values such as $d = 10$ and $d = 20$ was to be able to compare the performance of our algorithm with those in the literature. Unfortunately, for $d$ as large as 1000, the algorithms in the literature other than OD-LinBAI become infeasible due to their high computational complexity. In contrast, our algorithm is still computationally feasible for large $d$ (e.g., $d=500$). We added a synthetic data experiment with $d = 500$ to address this issue. This can be found in Section 5.2. Please also see the comparison of the runtimes of various algorithms in the newly created Table 2.
>
> 3. It is true that the sparse FC setting is as interesting as the sparse FB setting and that the complexity of the state-of-the-art algorithm (ALBA) from Tao et al. 2018 depends on the dimension $d$. Specifically, it is $O(\sum_{i = 2}^d \frac{1}{\Delta_i^2} \ln \frac{1}{\delta})$. It could be possible to improve the dependence on $d$ to a function of $s$ if we combined with the thresholded Lasso (with a careful optimization of the hyperparameters). However, experimentally, we observed that the performance dependence on the dimension in the FC setting is much weaker than that in the FB setting, which is also apparent in Tao et al.'s experiments in Figure 1 of their paper. Therefore, we decided not to study the FC setting in this paper. As it is, the discussion of the FB setting has taken up all of the given pages; hence, we leave the design and analysis of FC sparse BAI to future work.
>
> Requested Changes:
>
> 1. We added a real-world data experiment with $d = 55$ and a synthetic data experiment with $d = 500$. This can be found in Section 5.
>
> 2. Please see our response above.
>
> 3. We re-wrote the section after Corollary 1 to describe the parameter choices.

---

### Decision · Action_Editor_rdYt · 2024-01-19

**Recommendation:** Accept as is

**Comment:**

The reviewers are in favor of accepting this paper:

- The theoretical results are new, although the reviewers  think that the minimum signal condition somewhat makes the analysis straightforward.

- The original experiments and the additional high-dimensional / real-world data experiments makes the experimental evaluation convincing overall.

- In the camera-ready version, please take into account an additional technical comment below:

The presentation of the results needs more clarification. The reviewers and AE agree that that c_0 (19) in Corollary 1 can sometimes be O(1), but it seems not true for the standard basis action set example mentioned below (19) and in the rebuttal -- in that example isn't \phi^2 = \phi^2(1/d*I_d, s), and the best choice of \phi_0 is 1/d?

In defense of the paper's result, one can alternatively consider the hypercube action set A = {-1,+1}^d, in which case both x_max and \phi^2 are constant.

In view of this, we suggest that the authors provide detailed discussions on when c_0 is O(1), perhaps by supplying multiple examples.

**Audience:**

Yes

**Claims And Evidence:**

Yes